# Fairness-Aware Meta-Learning via Nash Bargaining

Yi Zeng[*1], Xuelin Yang[*2], Li Chen[3], Cristian Canton Ferrer[3], Ming Jin[1],
Michael I. Jordan[2], Ruoxi Jia[1]

[1]Virginia Tech, Blacksburg, VA 24061, USA
[2]University of California, Berkeley, CA 94720, USA
[3]Meta AI, Menlo Park, CA 94025, USA

## Abstract

To address issues of group-level fairness in machine learning, it is natural to adjust model parameters based on specific fairness objectives over a sensitive-attributed validation set. Such an adjustment procedure can be cast within a meta-learning framework. However, naive integration of fairness goals via meta-learning can cause hypergradient conflicts for subgroups, resulting in unstable convergence and compromising model performance and fairness. To navigate this issue, we frame the resolution of hypergradient conflicts as a multi-player cooperative bargaining game. We introduce a two-stage meta-learning framework in which the first stage involves the use of a *Nash Bargaining Solution* (NBS) to resolve hypergradient conflicts and steer the model toward the Pareto front, and the second stage optimizes with respect to specific fairness goals. Our method is supported by theoretical results, notably a proof of the NBS for gradient aggregation free from linear independence assumptions, a proof of Pareto improvement, and a proof of monotonic improvement in validation loss. We also show empirical effects across various fairness objectives in six key fairness datasets and two image classification tasks.

## 1 Introduction

The traditional formulation of machine learning is in terms of a system that improves its predictive and decision-making performance by interacting with an environment. Such a formulation is overly narrow in emerging applications—it lumps the social context of a learning system into the undifferentiated concept of an "environment" and provides no special consideration of the collective nature of learning. Such social context includes notions of scarcity and conflict, as well as goals such as social norms and collaborative work that are best formulated at the level of social collectives. The neglect of such considerations in traditional machine learning leads to undesirable outcomes in real-world deployments of machine learning systems, including outcomes that favor particular groups of people over others [44, 7, 31, 10, 38, 51], the amplification of social biases and stereotypes [28, 14, 47], and an ongoing lack of clarity regarding issues of communication, trust, and fairness.

Our focus of the current paper is fairness, and we take a perspective on fairness that blends learning methodology with economic mechanisms. The current favored methodology for addressing fairness recognizes that it is not a one-size-fits-all concept—different fairness notions are appropriate for different social settings [49, 32, 50]—and treats fairness via meta-learning ideas. Meta-learning is implemented algorithmically with the tools of bi-level optimization. Specifically, fairness-aware meta-learning employs outer optimization to align with a specific fairness goal over a small, demo-

---

[*]Xuelin and Yi contributed equally. Yi's work is partially done at Responsible AI, Meta AI. Corresponding: Li Chen and Ruoxi Jia. Code: Nash-Meta-Learning.

38th Conference on Neural Information Processing Systems (NeurIPS 2024).

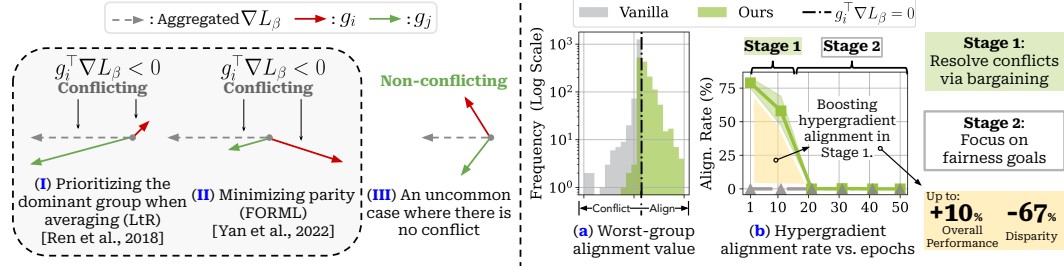

Figure 1: **Overview:** We illustrate the problem of *hypergradient conflicts* in conventional one-stage fairness-aware meta-learning, which we find can lead to erratic performance and/or convergence at suboptimal, unfair local minima. **Left:** Graphical depiction of group-wise hypergradient conflicts (**I**, **II**) showing different scenarios where conflicts arise in one-stage meta-learning, affecting performance stability; (**III**) provides a depiction of the contrast case where the aggregated direction is not conflicting with any of the groups, which leads to a more stable, fair, and performant model. **Right:** (**a**, **b**) Comparison of conventional one-stage meta-learning (Vanilla showcased by FORML, highlighted in gray) with our proposed two-stage meta-learning approach (Ours, highlighted in green) which resolves hypergradient conflicts through a bargaining process. In our evaluation, we show the efficacy of our method in enhancing fairness-aware meta-learning, with improvements in performance by up to 10% and fairness by up to 67%, by initially focusing on conflict resolution in Stage 1 to steer the model towards the Pareto front followed by focusing on fairness goals in Stage 2.

graphically balanced validation set to adjust a set of hyperparameters, while the inner optimization minimizes the hyperparameter-adjusted training loss [43, 52, 53]. This approach addresses two central challenges in group-level algorithmic fairness. First, it can integrate distinct fairness goals into the outer optimization. This flexibility allows customization of the focus of objectives, including enhancing the averaged loss across demographic- and label-balanced groups [43] and minimizing disparities [53]. This is a conceptual improvement over methods confined to a single fairness objective [18, 27]. Additionally, meta-learning reduces the reliance on sensitive attribute labels in the training data. This circumvents the label dependence in conventional methods [30, 22, 21] and addresses ethical concerns over the acquisition of sensitive attributes [1].

Although these arguments suggest that a meta-learning approach is promising for group-level fairness, it stops short of providing an economic mechanism which embodies fairness in terms of allocations and the management of conflict. Our work aims to bridge this gap by bringing a concept from economic mechanism design—that of *Nash bargaining*—into contact with meta-learning. Specifically, in our initial empirical explorations of meta-learning algorithms, we found that performance and fairness can vary substantially according to the choice of fairness metric across different datasets, suggesting a form of conflict that is not being resolved effectively via basic meta-learning procedures. Investigating further, we identified a phenomenon that we term *hypergradient conflict* which we believe is a pivotal factor in driving the contrast in effectiveness among different fairness goals when integrated with meta-learning. Briefly, the aggregated gradient of the outer optimization objective (the *hypergradient*) conflicts with the desired update associated with particular groups (Figure 1). To address this, we propose a novel framework that resolves hypergradient conflicts as a cooperative bargaining game. Specifically, we present a two-stage meta-learning framework for fairness: first incorporate the *Nash Bargaining Solution* (NBS) at an early training stage to mitigate conflicts and steer the model toward the Pareto front, and then engage in the pursuit of specified fairness goals.

Our work also introduces a new derivation of the NBS for gradient aggregation, one that dispenses with gradient independence assumptions made in the past work so that it is applicable to broader contexts. This derivation may be of independent interest, and accordingly we present material on game-theoretic justification, Pareto improvement, and monotonic improvement in validation loss that help to relate the NBS to our gradient-based learning setting. Our analysis sheds light on the convergence exhibited in empirical studies and provides an understanding of how our method improves meta-learning for fairness.

| | Baseline | LtR | FORML | Meta-gDRO |
|---|---|---|---|---|
| **Adult Income** [3], (sensitive attribute: **race**) | | | | |
| **Overall AUC** (↑) | 0.668 | 0.803 (+20%) | 0.710 (+6.2%) | 0.775 (+16%) |
| **Max-gAUCD** (↓) | 0.225 | 0.090 (-96%) | 0.290 (+29%) | 0.163 (-28%) |
| **Worst-gAUC** (↑) | 0.544 | 0.755 (+38%) | 0.540 (-0.7%) | 0.694 (+28%) |
| **Titanic Survival** [12], (sensitive attribute: **sex**) | | | | |
| **Overall AUC** (↑) | 0.972 | 0.967 (-0.5%) | 0.950 (-2.3%) | 0.961 (-1.1%) |
| **Max-gAUCD** (↓) | 0.056 | 0.044 (-21%) | 0.033 (-41%) | 0.033 (-41%) |
| **Worst-gAUC** (↑) | 0.944 | 0.944 (-) | 0.933 (-1.2%) | 0.944 (-) |

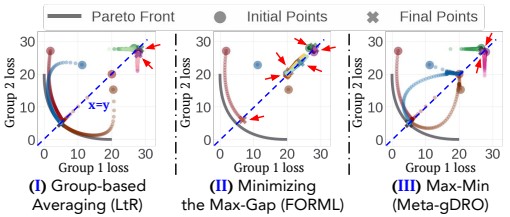

(a) Comparison of the model performance of three conventional one-stage meta-learning methods (LtR, FORML, and Meta-gDRO) across various fairness notions (results averaged from 5 runs, % relative to the baseline only using the training set w/o meta-learning)

(b) Trajectory of 1000-step optimizations. X and y-axis: validation losses for groups 1 and 2, rsp. Gray curves (lower left): the Pareto front. **x=y**: fairness (equal validation loss). Ideal endpoint: their intersections. Undesirable endpoints: "➡". 6 random initials.

Figure 2: The unreliable performance of conventional one-stage fairness-aware meta-learning.

Finally, we present a thoroughgoing set of empirical studies that evaluate our method using synthetic and real-world datasets, encompassing six key fairness datasets and two image classification tasks. As we'll show, our framework uniformly enhances the performance of one-stage meta-learning methods, yielding up to 10% overall performance improvement and a 67% drop in disparity (Figure 1).

The remainder of this paper is structured as follows: Section 2 describes the problem. Section 3 details our method, including the bargaining game formulation, solution, and theoretical analyses (§3.1-3.3), and our two-stage meta-learning framework, its theoretical foundations, and dynamical issues (§3.4-3.6). Section 4 presents empirical results and analysis. Related work is deferred to Appendix A.1.

## 2 Problem Statement

Let $\theta$ denote a vector of model parameters, let $D^{(train)}$ denote the training data and let $D^{(val)}$ a validation set, with $K$ sensitive groups $D_i^{(val)}$, for $i \in [K]$. Let $L$ be a vector-valued function where each entry corresponds to the per-example conventional loss (e.g., cross-entropy). Define *group-wise validation loss* $L^{(val)}$ as a vector of size $K$ where $L_i^{(val)} = \frac{1}{|D_i^{(val)}|} L(D_i^{(val)}|\theta^*(w))^\top \mathbb{1}$

(the averaged loss over samples in group $i$). Define $L_\beta = \beta^\top L^{(val)}$ as the fairness loss with vector $\beta$ of size $K$ encoding the fairness objectives under consideration. Following established work [43, 53], we target group-level fairness objectives via meta-learning as follows:

$$w^* = \arg\min_{w \geq 0} L_\beta(D^{(val)}|\theta^*(w)), \tag{1}$$

$$\theta^*(w) = \arg\min_\theta w \cdot L(D^{(train)}|\theta). \tag{2}$$

The vector $w$ is a set of hyperparameters that reweigh each training example in the current minibatch when updating $\theta$, optimized on $D^{(val)}$ to improve group-level fairness.

Existing fairness-aware meta-learning work can be characterized as different protocols for $\beta$. **LtR** [43] computes the average of all group-losses with demographic and label balanced $D^{(val)}$ (i.e., $\beta_{\text{LtR}} = \frac{1}{K}\mathbb{1}$). **FORML** [53] calculates the difference between the maximum and minimum group-loss (i.e., $\beta_{\text{FORML}}$ has 1 for max, -1 for min, and 0 otherwise), emphasizing parity. Meanwhile, group-level Max-Min fairness inspired from [41] focuses solely on the maximum group-loss [44], yielding a procedure referred to as **Meta-gDRO**, with $\beta_{\text{Meta-gDRO}}$ set equal to one for the max and zero otherwise. Throughout the training process, the hypergradient of these methods, $\nabla_w L_\beta(D^{(val)}|\theta^*(w))$, is derived by applying the aggregation protocol $\beta$ to the *group-wise hypergradients*, $\nabla_w L_i^{(val)}(D^{(val)}|\theta^*(w))$. Their training process, wherein the aggregated hypergradient is iteratively utilized to update $w$, is referred to as a *one-stage method*. They differ from our *two-stage method*, which employs distinct hypergradient aggregation rules at two separate stages.

**Unreliability of one-stage meta-learning for fairness.** The traditional approach of plugging a fairness objective into $L_\beta$ seems natural. However, we find that the effect of this approach on performance and fairness can be unstable. We evaluate the above three one-stage fairness-aware

methods based on targeted fairness metrics (detailed settings in §4.2): Overall Area Under the Curve (**Overall AUC**) for LtR which also measures prediction performance, maximum group AUC disparity (**Max-gAUCD**) for FORML, and worst group AUC (**Worst-gAUC**) for Meta-gDRO (Figure 2a).

**Delving into the cause.** After training on real data for several epochs, we find that most subsequent epochs have *fewer than 3%* of the aggregated hypergradients aligned to the optimization objectives of each subgroup. That is, we see *hypergradient conflicts*:

$$\exists i, \text{ s.t. } g_i^\top \nabla L_\beta < 0, \tag{3}$$

with the group-wise gradient $g_i = \nabla_w L_i^{(val)}(D^{(val)}|\theta(w))$ and aggregated direction $\nabla L_\beta = \nabla_w L_\beta(D^{(val)}|\theta(w))$. The prevalence of intrinsic hypergradient conflict in one-stage methods is unsurprising, because their aggregation methods are unable to prevent overlooking or incorrectly de-prioritizing certain groups. We study this phenomenon in synthetic settings to isolate structural issues from randomness in stochastic optimization (Figure 2b, settings in Appendix A.6): We define the performance goal as convergence to the Pareto front, while the fairness goal corresponds to the line $x = y$. As observed, alignment issues in synthetic experiments are prominent (Figure 2b): (**I**) Averaging (LtR) may induce oscillatory dominance among groups. (**II**) Parity-based method (FORML) produces conflicting hypergradients as it subtracts the loss of one group, necessitating performance trade-offs for fairness. (**III**) Minimizing the worst-group loss (Meta-gDRO) often exhibits toggling dominance as it solely prioritizes the current least-performing group, which may create conflicts and cannot land on the Pareto front.

**Hypergradient conflict resolution.** Given the observation of hypergradient conflicts and convergence issues in one-stage meta-learning, we turn to cooperative bargaining and propose a two-stage method that seeks to attain more reliable improvements by resolving conflicts at the early stage of training. Our methodology draws inspiration from the *Nash Bargaining Solution* (NBS), a cornerstone of axiomatic bargaining in game theory, known for its general applicability and robustness [34]. Nash Bargaining is chosen for its desirable axiomatic properties, which prohibit unconsented unilateral gains by Pareto Optimality, and its principled approach of effectively balancing interests, making it appealing for practical deployment. We provide additional discussion of the game-theoretic perspective and additional empirical comparisons in Appendix A.2.

While the NBS has been studied in multi-task learning [37], it has yet to be explored in fairness-aware meta-learning. Unlike the settings in [37], applying the NBS in our context challenges the assumption of linear independence among tasks, which is generally untenable for group-wise utility towards the same goal of performance gain (i.e., the settings of fairness). This drives our exploration into novel proofs and applications of the NBS in hypergradient aggregation, aiming to circumvent the need for linear independence and optimize shared outcomes through strategic negotiation and nested optimization.

## 3 Methodology

### 3.1 Nash Bargaining framework

We start with some preliminaries. Consider $K$ players faced with a set $A$ of alternatives. If all players reach a consensus on a specific alternative, denoted as $a$ in set $A$, then $a$ will be the designated outcome. In the event of a failure to agree on an outcome, a predetermined disagreement result, denoted as $d$, will be the final outcome. The individual utility payoff functions are denoted $u_i : A \cup \{D\} \to \mathbb{R}$, which represent the players' preferences over $A$. Denote the set of feasible utility payoffs as $S = (u_1(a), ..., u_K(a)) : a \in A \subset \mathbb{R}^K$ and the disagreement point as $d = (u_1(D), ..., u_K(D))$. Nash proposed to study solutions to the bargaining problem through functions $f : (S, d) \to \mathbb{R}$. The unique Nash bargaining solution (NBS), originally proposed for two players [34] and latter extended to multiple players [15], maximizes the function $f(S, d) = \prod_i (x_i - d_i)$, where $x_i$ is the bargained payoff and $d_i$ is the disagreement payoff for player $i$. The NBS fulfills four axioms: Pareto Optimality, Symmetry, Independence of Irrelevant Alternatives, and Invariant to Affine Transformations. See Appendix A.3 for detailed definitions, and A.4 for additional assumptions.

In our problem, we want to find $\tilde{w}$ in Algorithm 1, an intermediate vector for the optimal $w^*$ to reweigh each training sample. Let $L_i^{(val)}$ be the validation loss and $g_i = \nabla_{\tilde{w}} L_i^{(val)}$ be the hypergradient of group $i$. Let $G$ be the matrix with columns $g_i$. The central question is to find the protocol $\alpha$ as the

weights applied to individual group $i$'s loss. It associates an update step $\nabla L_\alpha$ to $\tilde{w}$ that improves the aggregated validation loss among all groups.

We frame this problem as a cooperative bargaining game between the $K$ groups. Define the utility function for group $i$ as

$$u_i(\nabla L_\alpha) = g_i^\top \nabla L_\alpha. \tag{4}$$

The intuition is that the utility tells us how much of proposed update is applied in the direction of hypergradient of group $i$ (as it can be written as $\|g_i\|\|\nabla L_\alpha\|\cos\delta$ with angle $\delta$ between $g_i$ and $\nabla L_\alpha$). This gives the projection of $\nabla L_\alpha$ along $g_i$, or the "in effect" update for group $i$. If $g_i$ and $\nabla L_\alpha$ aligns well, the utility of group $i$ is large (or *vice versa*). Denote $B_\epsilon$ the ball of radius $\epsilon$ centered at 0. We are interested in the update $\nabla L_\alpha$ in the agreement alternative set $A = \{\nabla L : \nabla L \in B_\epsilon, \nabla L_\alpha^\top g_i - D^\top g_i > 0, \forall i \in [K]\}$. Assume $A$ is feasible and the disagreement point is $D = 0$ (i.e. the update $\nabla L_\alpha = 0$, staying at $\tilde{w}$). The goal is to find a $\nabla L_\alpha$ that maximizes the product of the deviations of each group's payoff from their disagreement point. Since $u_i$ forms a (shifted) linear approximation at $\tilde{w}$, we are essentially maximizing the utility of $L_\alpha$ locally. We provide further discussions on the problem setup and assumptions in Appendix A.4.

## 3.2 Solving the problem

In this section, we will show the NBS is (up to scaling) achieved at $\nabla L_\alpha = \sum_{i \in [K]} \alpha_i g_i$, where $\alpha \in \mathbb{R}_+^K$ solves $G^\top G \alpha = \frac{1}{\alpha}$ by the following two theorems, with full proofs available in Appendix A.5:

**Theorem 3.1.** *Under $D = 0$, $\arg\max_{\nabla L_\alpha \in A} \prod_{i \in [K]}(u_i(\nabla L_\alpha) - d_i)$ is achieved at*

$$\sum_{i \in [K]} \frac{1}{\nabla L_\alpha^\top g_i} g_i = \gamma \nabla L_\alpha, \quad \text{for some } \gamma > 0. \tag{5}$$

**Proof Sketch.** We employ the same techniques as in Claim 3.1 of [37].

**Theorem 3.2.** *The solution to Equation 5 is (up to scaling) $\nabla L_\alpha = \sum_{i \in K} \alpha_i g_i$ where*

$$G^\top G \alpha = \frac{1}{\alpha} \tag{6}$$

*with the element-wise reciprocal $\frac{1}{\alpha}$.*

**Proof sketch.** Let $x = \nabla L_\alpha$. In line with [37] we observe that $x = \frac{1}{\gamma} \sum_{i \in K}(x^\top g_i)^{-1} g_i$. However, whereas [37] relied on the linear independence of the $g_i$'s to uniquely determine each coefficient $(x^\top g_i)^{-1}$, our technique makes no such assumption. Instead, we multiply both sides of Equation 5 by $g_j$ and obtain $\sum_{i \in [K]}(x^\top g_i)^{-1}(g_i^\top g_j) = \gamma x^\top g_i, j \in [K]$. Set $\alpha_i = (x^\top g_i)^{-1}$. This is equivalent to $g_j^\top \sum_{i \in [K]} g_i \alpha_i = \alpha_j^{-1}$, which is the desired solution when written in the matrix form.

While our solution aligns with that of multi-task learning [37], our proof of Theorem 3.2 circumvents the necessity for linear independence among $g_i$, one of the core initial assumptions in the previous work. Linear independence does not hold in general as the goals for individual groups (or tasks) might overlap (such as sharing common underlying features) or contradict each other (when there is negative multiplicity). Our proof removes this assumption and sheds light on the effectiveness of updates based on the NBS in general cases. See Appendix A.5 for extended discussions on linear independence.

Furthermore, we derive two useful properties of the NBS in additional to its four axioms, with full proofs available in Appendix A.5:

**Corollary 3.3.** *(Norm of bargained update) The solution in Theorem 3.2 has $\ell^2$-norm $\sqrt{K}$.*

**Corollary 3.4.** *If $g_j$ is $\sigma$-bounded for $j \in [K]$, $\|\alpha_j^{-1}\|$ is $(\sqrt{K}\sigma)$-bounded for $j \in [K]$.*

Informally, the NBS has implicit $\ell^2$ regularization (Corollary 3.3) which substantiates our empirical observation that a separate $\ell^2$-normalization on $\nabla L_\alpha$ for meta-learning rate adjustment yields better performance than $\ell^1$-normalization (the conventional setting in [43]). Note that we do not impose any assumptions on the boundedness of the hypergradient $g_i$, ensuring the stability even when certain hypergradients are extreme. Furthermore, when $g_i$ is bounded, Corollary 3.4 implies that $\|\alpha_j\|$ is bounded below and no groups are left behind.

**Algorithm 1:** Two-stage Nash-Meta-Learning Training

| | |
|---|---|
| **Input:** | $\theta^{(0)}$; $D^{(train)}$; $D^{(val)} = \{D_1^{(val)}, \ldots, D_K^{(val)}\}$ |
| **Parameters:** | $\beta_0$ (fairness protocol); $\eta^{(t)}$ (step size); |
| | $T_{bar}$ (bargaining steps) |

**1** **for** *step* $t \in \{1, \ldots, T\}$ **do**
**2**     Minibatch $D^{(t)}$ sampled from $D^{(train)}$;
    /* 1. Unrolling Inner Optimization */
**3**     $\nabla\theta^{(t)}(\tilde{w}) \leftarrow \textbf{BackwardAD}_\theta\big(\tilde{w} \cdot L(D^{(t)}|\theta^{(t)})\big)$;
**4**     $\hat{\theta}^{(t)}(\tilde{w}) \leftarrow \theta^{(t)} - \eta^{(t)}\nabla\theta^{(t)}(\tilde{w})$;
**5**     $\beta \leftarrow \beta_0$;
**6**     **if** $t < T_{bar}$ **then**
       /* 2. Validation Group-Utilities */
**7**        **for** *group* $k \in \{1, \ldots, K\}$ **do**
**8**           $g_k^{(t)} \leftarrow \textbf{BackwardAD}_{\tilde{w}}\big(L_k^{(val)}(D^{(val)}|\hat{\theta}^{(t)}(\tilde{w}))\big)$;
       /* 3. Bargaining Game */
**9**        Set $G^{(t)}$ with columns $g_k^{(t)}$;
**10**       Solve for $\alpha$: $(G^{(t)})^T G^{(t)}\alpha = \frac{1}{\alpha}$ to obtain $\alpha^{(t)}$;
**11**       **if** $\alpha^{(t)}$ *is not None* **then**
          // `bargaining succeeded`
**12**           $\beta \leftarrow \alpha^{(t)}$;
**13**     $\tilde{w} \leftarrow \textbf{BackwardAD}_{\tilde{w}}\big(L_\beta(D^{(val)}|\hat{\theta}^{(t)}(\tilde{w}))\big)$;
    /* 4. Weighted Update */
**14**     $w \leftarrow \textbf{Normalize}\big(\max(-\tilde{w}, 0)\big)$;
**15**     $\nabla\theta^{(t)} \leftarrow \textbf{BackwardAD}_\theta\big(w \cdot L(D^{(t)}|\theta^{(t)})\big)$;
**16**     $\theta^{(t+1)} \leftarrow \theta^{(t)} - \eta^{(t)}\nabla\theta^{(t)}$;
**Output:** $\theta^{(T)}$

## 3.3 Game-theoretic underpinnings

The NBS provides incentives for each player to participate in the bargaining game by assuming the existence of at least one feasible solution that all players prefer over disagreement ($\exists x \in S$ s.t. $x \succ d$). This aligns players' interests in reaching an agreement. The constraint $g_i^\top \nabla L_\alpha > 0$ resolves hypergradient conflict upon agreement.

Second, note that Equation 6 shows relationship between the individual and interactive components:

$$\|\alpha_i g_i\|_2^2 + \sum_{j \neq i}(\alpha_i g_i)^\top (\alpha_j g_j) = 1, \tag{7}$$

for $i \in [K]$. [†] The relative weights $\alpha_i$ emerge from a player's own impact ($\|\alpha_i g_i\|_2^2$) and interactions with others ($(\alpha_i g_i)^\top(\alpha_j g_j)$). This trade-off embodies individual versus collective rationality. Positive interactions (i.e., $g_i^\top g_j > 0$) incentivize collective improvements by downweighting $\alpha_i$. Negative interactions (i.e., $g_i^\top g_j < 0$) increase $\alpha_i$ to prioritize individual objectives. Furthermore, each player accounts for a nontrivial contribution to the chosen alternative $\nabla L_\alpha$ under mild assumptions (Corollary 3.4). The negotiated solution balances individual and collective rationality through participation incentives and conflict resolution. This equilibrium encapsulates game-theoretic bargaining.

## 3.4 Two-stage Nash-Meta-Learning

We now present our two-stage method (Algorithm 1) that incorporates Nash bargaining into meta-learning training. Previous one-stage algorithms fix a predetermined $L_\beta$; our two-stage method assigns $\beta = \alpha$ in the NBS in Stage 1 and sets it back to the original $\beta_0$ in Stage 2. Define $\theta^{(t)}$ as the

---

[†]To give game-theoretic justifications, we build on a similar expression (Equation 2, [37]).

model parameter at step $t$, $T$ as the number of total steps, and $T_{(bar)} \leq T$ as the number of bargaining steps in Stage 1. Let **BackwardAD**$_\theta(L)$ be the backward automatic differentiation of computational graph $L$ w.r.t. $\theta$, and **Normalize**$(\cdot)$ be $\ell^2$-normalization function. Each step $t$ is constituted by four parts: **The first part** is unrolling inner optimization, a common technique to approximate the solution to a nested optimization problem [17]. We compute a temporary (unweighted) update $\hat{\theta}^{(t)}$ on training data, which will be withdrawn after obtaining the updated $w$. $\tilde{w}$ is initialized to zero and included into the computation graph. **The second part** calculates the hypergradient $g_k^{(t)}$ which ascertains the descent direction of each training data locally on the validation loss surface. **The third part** is to aggregate the hypergradients as the update direction for $\tilde{w}$ by $\alpha$ from the NBS. In Stage 1, successful bargaining grant the update of $\tilde{w}$ by the NBS. If the bargaining game is infeasible or if we are in Stage 2 (not in the bargaining steps), we calculate $\tilde{w}$ based on the fairness loss $L_\beta$ of our choice. **The last part** is the update of parameter $\theta^{(t+1)}$ using the clipped and normalized weights $w$ for each training data in the minibatch.

## 3.5 Theoretical Properties

**Theorem 3.5.** *(Update rule of $\theta$) Denote $L_i^{(train)} = L(D_i^{(t)}|\theta^{(t)}) \in \mathbb{R}$ for the $i$-th sample in training minibatch $D^{(t)}$ at step $t$. $\theta$ is updated as $\theta^{(t+1)} = \theta^{(t)} - \frac{\eta^{(t)}}{|D^{(t)}|} \sum_{i=1}^{|D^{(t)}|} \Delta\theta^i$ with $\Delta\theta^i = \max\left(\left(\nabla_\theta((\beta^{(t)})^\top L^{(val)})\right)^\top \nabla_\theta L_i^{(train)}, 0\right) \nabla_\theta L_i^{(train)}$.*

**Theorem 3.6.** *(Pareto improvement of $\tilde{w}$) Use $\alpha^{(t)}$ for the update. Assume $L_i^{(val)}$ is Lipschitz-smooth with constant $C$ and $g_i^{(t)}$ is $\sigma$-bounded at step $t$. If the meta learning rate for $\tilde{w}$ satisfies $\eta^{(t)} \leq \frac{2}{CK\alpha_j^{(t)}}$ for $j \in [K]$, then $L_i^{(val)}(\tilde{w}^{(t+1)}) \leq L_i^{(val)}(\tilde{w}^{(t)})$ for any group $i \in [K]$.*

**Theorem 3.7.** *Assume $L^{(val)}$ is Lipschitz-smooth with constant $C$ and $\nabla_\theta L_i^{(train)}$ is $\sigma$-bounded. If the learning rate for $\theta$ satisfies $\eta^{(t)} \leq \frac{2|D^{(t)}|}{C\|\beta^{(t)}\|\sigma^2}$, then $L_{\beta^{(t)}}(\theta^{(t+1)}) \leq L_{\beta^{(t)}}(\theta^{(t)})$ for any fixed vector $\beta^{(t)}$ with finite $\|\beta^{(t)}\|$ used to update $\theta^{(t)}$.*

Informally, the closed-form update rule of $\theta$ indicates that the weight of a training sample is determined by its local similarity between the $\beta$-reweighed validation loss surface and the training loss surface (Theorem 3.5). Under mild conditions, $\tilde{w}$ yields Pareto improvement for all groups for the outer optimization using the NBS (Theorem 3.6). For the inner optimization, under mild conditions, the fairness loss $L_{\beta^{(t)}}$ monotonically decreases w.r.t. $\beta^{(t)}$ regardless of the choice of protocol (Theorem 3.7). This generalizes [43] from $\beta = \frac{1}{K}\mathbb{1}$ to any $\beta$ with finite norm and provides a uniform property for the fairness-aware meta-learning methods. It entails the flexibility of our two-stage design that switches $\beta$ between phases. Setting $\beta = \alpha$ gives the desired property for the NBS. The validation loss surface reweighed by the NBS has the maximum joint utility, which empirically boosts the overall performance when used to update $\theta$. Full proofs are given in Appendix A.5.

## 3.6 Dynamics of Two-stage Nash-Meta-Learning

Our method captures the interplay and synergy among different groups. Specifically, previous methods like linear scalarization (i.e., assigning a fix weight to each group) are limited to identifying points on the convex envelope of the Pareto front [5]. Our method offers a more adaptable navigation mechanism by dynamically adjusting the weight with the NBS, which accounts for the intricate interactions and negotiations among groups. Moreover, the optimal $\nabla L_\alpha$ maximizes the utilization of information on validation loss landscape and leads to empirical faster and more robust convergence to the Pareto front even with distant initial points. Although first-order methods typically avoid saddle points [16, 39], if one is encountered, switching to the fairness goal upon unsuccessful bargaining offers a fresh starting point for subsequent bargaining iterations and helps to escape (Figure 6d, Appendix A.7). Our synthetic experiments show that Stage 2 training focused solely on the fairness goal does not deviate the model from the Pareto front (Figure 3). Specifically, the worst group utility $g_i \nabla L_\beta$ tends to concentrate around zero, ensuring the model stays in the neighborhood of the Pareto front and implying the robustness of our approach. The theoretical understanding of this interesting phenomenon is an open problem for future study.

# 4 Evaluation

We evaluate our method in three key areas: synthetic simulation (§4.1) for Pareto optimality vis-à-vis fairness objectives, real-world fairness datasets (§4.2), and two imbalanced image classification scenarios (Appendix A.7.3).

## 4.1 Simulation

Under the synthetic settings (§2 and Appendix A.6), we observe the convergence enhancements compared to Figure 2b and the effect of continuous bargaining throughout the entire training (another case of one-stage). Figure 3 demonstrates that Nash bargaining effectively resolves gradient conflicts and facilitates convergence to the Pareto front in comparison to Figure 2b. This is evident from the reduced number of non-Pareto-converged nodes in both continuous bargaining (Figure 3a) and early-stage bargaining (our final solution, Figure 3b). Notice that one-stage NBS doesn't always enhance fairness, evidenced by the observation that nodes at the Pareto front tend to stagnate. The NBS does not leverage any information about specific fairness objectives. Our two-stage approach built on the bargaining steps can further push the model to the ideal endpoints.

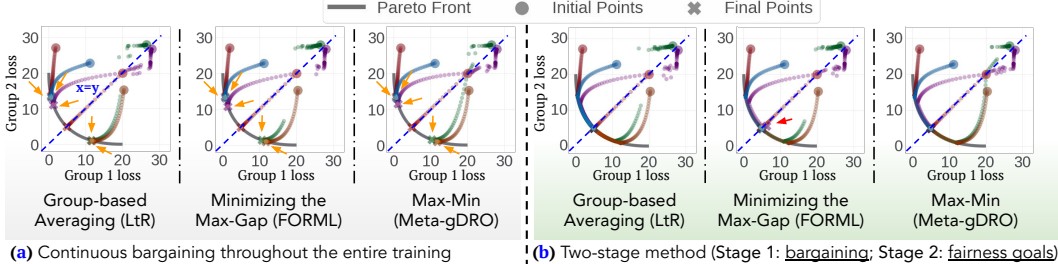

Figure 3: Synthetic illustration of the bargaining effects. "➡": final point not close to the fairness goal (x=y). "➡": final point not at the Pareto front. (a) Bargaining across all 1000 steps; (b) Bargaining only included in the first 100 steps (two-stage method).

## 4.2 Standard fairness benchmarks

We test our method on six standard fairness datasets across various sectors of fairness tasks: financial services (Adult Income [3], Credit Default [54]), marketing (Bank Telemarketing [33]), criminal justice (Communities and Crime [42]), education (Student Performance [11]), and disaster response (Titanic Survival [12]). Test sets comprise 3% of each dataset (10% for the student performance dataset with 649 samples) by randomly selecting a demographically and label-balanced subset. See Table 2 in Appendix A.6 for data distribution specifics.

**General settings and metrics.** We compare our two-stage Nash-Meta-Learning with conventional one-stage fairness-aware meta-learning (i.e., LtR, FORML, Meta-gDRO), baseline training, and Distributional Robust Optimization (DRO) [18]. All methods share the same model architecture and training hyperparameters on each dataset. Our approach features a 15-epoch bargaining phase within the total 50 epochs. See Appendix A.6 for training details. Unlike synthetic experiments, the Pareto Front of real world datasets could be computationally intractable, so we cannot directly evaluate regarding this. Three metrics from §2 are used: Overall AUC (↑), Max-gAUCD (↓), and Worst-gAUC (↑), corresponding to the goal of LtR, FORML, and Meta-gDRO, respectively.

**Results and analysis.** Our NBS-enhanced two-stage meta-learning improves the overall performance, fairness objectives (in color), and stability, as in Table 1, and with 95% CI in Table 4, Appendix A.7. While the results without bargaining majorly agree with the prior work, bargaining increases FORML's Overall AUC by 10.34% (from 0.706 to 0.779) with a tighter 95% CI (from 0.202 to 0.054), and decreases Max-gAUCD by 26% (from 0.039 to 0.029) with a tighter CI (from 0.046 to 0.018). However, our method faced challenges in two datasets: Credit Default, where performance and fairness occasionally declined, and Communities and Crime, where minimal improvement was observed (in particular, the Meta-gDRO). We diagnose that these two dataset's validation set contains low feature-label mutual information, leading to noisy outcomes (Appendix A.7) and affecting

| | Baseline | DRO | LtR | | FORML | | Meta-gDRO | |
|---|---|---|---|---|---|---|---|---|
| | | | one-stage | two-stage (ours) | one-stage | two-stage (ours) | one-stage | two-stage (ours) |
| **I. Adult Income** [3], (sensitive attribute: **sex**) | | | | | | | | |
| **Overall AUC** (↑) | 0.778 | 0.761 | 0.830 | 0.830 (+.000) | 0.801 | 0.810 (+.009) | 0.810 | **0.837** (+.027) |
| **Max-gAUCD** (↓) | **0.016** | 0.029 | 0.050 | 0.047 (-.003) | 0.075 | 0.031 (-.044) | 0.052 | 0.046 (-.006) |
| **Worst-gAUC** (↑) | 0.770 | 0.747 | 0.805 | 0.807 (+.002) | 0.763 | 0.795 (+.032) | 0.809 | **0.814** (+.006) |
| **II. Adult Income** [3], (sensitive attribute: **race**; * one group in training data contains only one positive (favorable) label. ) | | | | | | | | |
| **Overall AUC** (↑) | 0.668 | 0.652 | 0.803 | **0.805** (+.002) | 0.710 | 0.775 (+.065) | 0.775 | 0.793 (+.018) |
| **Max-gAUCD** (↓) | 0.225 | 0.236 | **0.090** | 0.090 (-.000) | 0.290 | 0.134 (-.156) | 0.163 | 0.158 (-.005) |
| **Worst-gAUC** (↑) | 0.544 | 0.538 | 0.755 | **0.760** (+.005) | 0.540 | 0.688 (+.148) | 0.694 | 0.703 (+.009) |
| **III. Bank Telemarketing** [33], (sensitive attribute: **age**) | | | | | | | | |
| **Overall AUC** (↑) | 0.697 | 0.686 | 0.724 | 0.728 (+.004) | 0.706 | **0.779** (+.073) | 0.698 | 0.722 (+.024) |
| **Max-gAUCD** (↓) | **0.013** | 0.025 | 0.099 | 0.083 (-.016) | 0.039 | 0.029 (-.010) | 0.098 | 0.079 (-.019) |
| **Worst-gAUC** (↑) | 0.691 | 0.691 | 0.675 | 0.686 (+.011) | 0.686 | **0.764** (+.078) | 0.649 | 0.683 (+.034) |
| **IV. Credit Default** [54], (sensitive attribute: **sex**; * val data is more noisy than others, see analysis in Table 5, Appendix A.7. ) | | | | | | | | |
| **Overall AUC** (↑) | 0.634 | 0.624 | 0.630 | 0.616 (-.014) | 0.554 | 0.611 (+.057) | **0.682** | 0.661 (-.021) |
| **Max-gAUCD** (↓) | 0.024 | 0.022 | 0.037 | 0.025 (-.012) | 0.017 | 0.033 (+.016) | **0.016** | 0.022 (+.006) |
| **Worst-gAUC** (↑) | 0.622 | 0.613 | 0.612 | 0.603 (-.009) | 0.545 | 0.595 (+.050) | **0.674** | 0.650 (-.024) |
| **V. Communities and Crime** [42], (sensitive attribute: **blackgt6pct**; * val data is noisy, meanwhile one group in training data are all positive labels ) | | | | | | | | |
| **Overall AUC** (↑) | 0.525 | 0.568 | 0.679 | **0.700** (+.021) | 0.554 | 0.568 (+.014) | 0.686 | 0.686 (+.000) |
| **Max-gAUCD** (↓) | **0.050** | 0.136 | 0.071 | 0.129 (+.058) | 0.107 | 0.136 (+.029) | 0.114 | 0.114 (-.000) |
| **Worst-gAUC** (↑) | 0.500 | 0.500 | **0.643** | 0.636 (-.007) | 0.500 | 0.500 (+.000) | 0.629 | 0.629 (+.000) |
| **VI. Titanic Survival** [12], (sensitive attribute: **sex**) | | | | | | | | |
| **Overall AUC** (↑) | 0.972 | **0.983** | 0.967 | 0.978 (+.011) | 0.950 | 0.972 (+.022) | 0.961 | 0.972 (+.011) |
| **Max-gAUCD** (↓) | 0.056 | 0.033 | 0.044 | 0.044 (-0.00) | 0.033 | **0.011** (-.022) | 0.033 | 0.033 (-.000) |
| **Worst-gAUC** (↑) | 0.944 | **0.967** | 0.944 | 0.956 (+.012) | 0.933 | **0.967** (+.034) | 0.944 | 0.956 (+.012) |
| **VII. Student Performance** [11], (sensitive attribute: **sex**) | | | | | | | | |
| **Overall AUC** (↑) | 0.784 | 0.816 | 0.900 | 0.900 (+.000) | 0.828 | 0.822 (-.006) | 0.909 | **0.912** (+.003) |
| **Max-gAUCD** (↓) | 0.119 | 0.106 | **0.013** | 0.037 (+.024) | 0.056 | 0.031 (-.025) | 0.031 | 0.025 (-.006) |
| **Worst-gAUC** (↑) | 0.725 | 0.762 | 0.894 | 0.881 (+.013) | 0.800 | 0.806 (+.006) | 0.894 | **0.900** (+.006) |

Table 1: Comparison on standard fairness datasets (averaged from 5 runs). Each of { LtR , FORML , Meta-gDRO } is paired with { Overall AUC , Max-gAUCD , Worst-gAUC } rsp. for aligning intended fairness goals. Top results of each row in **bold**.

most tested methods (e.g., LtR, FORML). Additionally, for Communities and Crime, our method is influenced by the low bargaining success/feasible rate, possibly due to the lack of favorable (positive) training samples for the minority groups (Table 2, Appendix A.6). Conversely, our method still yields the anticipated bargaining results on the adult income dataset with only one positive `Amer-Indian` sample. These insights emphasize the importance of validation set quality and representative samples in the training.

**Effects of bargaining on hypergradient conflicts.** Bargaining enhances hypergradient alignment by varying degrees among different one-stage algorithms (Figure 4, 6). For instance, LtR's alignment rate improves from 60% to 80%, and FORML jumps from 0 to 76.9% accompanied with more substantial performance and fairness gains on Bank Telemarketing (Figure 4). FORML consistently benefits more from bargaining compared to the other two, likely due to its optimization goal that could intensify hypergradient conflicts. Moreover, our approach uniformly promotes hypergradient alignment during Stage 1 (Figure 6). We show that early-stage bargaining, accompanied by its hypergradient conflict resolu-

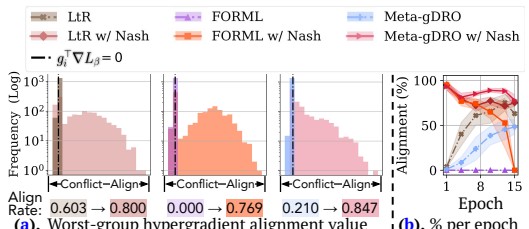

Figure 4: Effects on hypergradient alignment (Bank Telemarketing). (**a**) Smallest $g_i^\top \nabla L_\beta$. Portion of positive values (Align. Rate) at the bottom. (**b**) Hypergradient alignment rate per epoch.

tion, is crucial for enhancing model performance and fairness. Further illustrations and analyses are in Appendix A.7.

**Discussions on scalability.** To scale up models and datasets, one natural question is whether the bargaining game can still be feasible. For larger number of groups, although the likelihood of getting unresolvable conflict may get higher due to the fact that more players are getting involved, we

conjecture that the feasibility of $A$ depends more on the group structure rather than the number of groups. For example, the interdependencies and shared factors between groups may cause dependency in hypergradients to enable the feasibility of $A$. When goals of groups rely on common resources or have shared objectives at a higher level, it is still likely to have nonempty $A$.

## 5 Discussion & Conclusion

In conclusion, our study offers several key insights: We identified *hypergradient conflict* as a pivotal issue undermining stable performance in one-stage fairness-aware meta-learning algorithms. To mitigate this, we proposed a two-stage framework that initially employs the NBS to resolve these conflicts, and then optimizes for fairness. Our assumption-free proof of the NBS extended its applicability to a broader range of gradient aggregation problems. Empirical results demonstrated our method's superior performance across synthetic scenarios, seven real-world fairness settings on six key fairness datasets, and two image classification tasks.

**Future directions.** First, addressing the absence of a specific label in the training subgroup and the low quality of the validation set that affected our method's effectiveness may be mitigated by fairness-aware synthetic data [48] or data-sifting methods [56]. Pairing our method with them and exploring more resilient solutions adapted to these extreme cases can be a promising direction. Second, Theorem 3.7 establishes the validity to switch the choice of $\beta$ during training. Future work can focus on designing and flexibly choosing outer optimization goals to delicately improve performance, fairness, or other metrics in interest. Third, we derive the NBS under $D = 0$ for conflict resolution. Future study could investigate general $D \neq 0$, which might not be useful for conflict resolution (the scope of our paper), but could be used in other cases as a gradient aggregation method that gains advantages from axiomatic properties, as discussed in Appendix A.4.

## Acknowledgments

We acknowledge Aram H. Markosyan and Vitor Albiero at Meta AI for their insightful discussions. Ruoxi Jia and the ReDS lab acknowledge the support from the National Science Foundation under Grant No. IIS-2312794. Ming Jin acknowledges the support from the National Science Foundation under Grants ECCS-2331775 and IIS-2312794. Michael Jordan wishes to acknowledge support from the European Union (ERC-2022-SYG-OCEAN-101071601).

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

# A  Appendix

In this section, we first provide background and related work (§A.1), along with context on the motivation of choosing the NBS (§A.2). Then, we give formal definitions of some preliminaries mentioned in the main text (§A.3). We supplement discussions on the gargaining game setup and assumptions (§A.4), followed by the full proofs of corollaries and theorems (§A.5). We then provide detailed experimental settings (§A.6) and the additional results and analysis on with 95% CI, test noise, gradient conflict resolution, and experiments on the imbalanced image classification task (§A.7). Finally, we discuss the broader impact and limitations (§A.8).

## A.1  Background & Related Work

**Bargaining game in ML.**    Bargaining has been broadly applied across various ML contexts such as multi-task learning [37], multi-agent reinforcement learning [46, 40, 9, 45], multi-armed bandits [2], feature selection [23], and Bayesian optimization [4]. For bargaining-related gradient aggregation, a key distinction between our work and prior research is our provision of a proof for the NBS that does not rely on the linear independence assumption, and we further extend from the basic gradient descent setting to bi-level fairness-aware meta-learning [37]. In the context of meta-learning for fairness, we critically examine and build upon the work of [43], as detailed in §2. To the best of our knowledge, we are the first work to incorporate Nash bargaining into fairness-aware meta-learning.

**Bias mitigation.**    Traditional bias mitigation methods such as relabeling [30], resampling [21], or reweighting [22] hinge upon the availability of sensitive attributes in training data. However, such attributes can often be inaccurate, incomplete, or entirely unavailable due to privacy and ethical concerns in the collection process [1]. In response, researchers have developed two lines of proxy-based strategies over the past decades. The first line of work adopts indirect features associated with the sensitive feature of interest, such as zip codes for ethnicity [13] or sound pitch for gender [26]. Despite circumventing the requirement of sensitive attributes in training data, the effectiveness of these proxies critically hinges on their correlation with the actual sensitive attributes. The second line of work aims at aligning with the Max-Min fairness principle and uses the worst-performing samples as the proxy of the most disadvantaged groups [41, 18, 27, 8]. Challenges also arise in the potential bias toward mislabeled data when targeting worst-performing samples [51]. Most importantly, with the proxy group only aligned with Max-Min fairness, these lines of mitigation cannot be generalized or adapted to help other fairness notions.

**Scope and Notions of Fairness.**    Our exploration of fairness is specifically tailored to group-level notions, as fairness-aware meta-learning inherently relies on a representative set of samples organized by group information for model updates. While Max-Min fairness was chosen as a common evaluation metric, our paper also encompasses two additional mainstream fairness notions, including demographic parity [53] and average performance across equally represented groups [43], both used as one-stage baselines (Section 2). Our proposed two-stage method is also evaluated on its effectiveness in reducing disparity and improving the overall performance compared to these two baselines, respectively, as shown in Table 1.

**Max-Min Fairness and Rawlsian Justice.**    Group-level Max-Min fairness is a concept originated from Rawl's definition of fairness or, equivalently, justice [41]. Rawls defines the least advantaged group by primary goods with objective characteristics, which are independent of specific predictors. The Difference Principle in Rawlsian Justice requires that the existing mechanism always contributes to the least advantaged group. While Rawlsian Justice has been extended to specific utilities in the context of ML group-level fairness [18, 27, 51], the worst-performing group in ML fairness may vary over epochs and depend on the optimization status, unlike the least advantaged group in the original Rawlsian context. Consequently, the group-level Max-Min fairness approach in ML may not necessarily create a safety net for the least advantaged group as the original Rawlsian Justice intends. Rather, the ML group-level Max-Min fairness provides a dynamic optimization strategy that maximizes the minimum performance across all groups at each epoch. This approach ensures that the worst-performing group, which may change over time, is prioritized during the optimization process. While this interpretation of fairness differs from the original Rawlsian context, it remains a valuable technique for promoting equity in ML systems by preventing any group from being consistently disadvantaged.

## A.2   Additional Discussions

**Why NBS for hypergradient conflict resolution? – A game theory perspective.** Game theory offers two main categories: cooperative and noncooperative games. Cooperative games involve players forming alliances to achieve common objectives, while noncooperative games (e.g., Nash equilibrium [36]) focus on players acting independently without contracts. Our problem aligns better with the cooperative category, where players collaborate to maximize their collective gains from the proposed hypergradient update. Cooperative bargaining, a subset of cooperative games, studies how players with distinct interests negotiate to reach mutually beneficial agreements, directly corresponding to our goal of conflict resolution. Among cooperative bargaining solutions, the NBS stands out for its general applicability, robustness [34], and unique solution satisfying desirable axioms including Pareto Optimality, Symmetry, Invariance to Affine Transformations, and Independence of Irrelevant Alternatives [35]. These properties make the NBS an attractive choice for resolving hypergradient conflicts in a principled and fair manner.

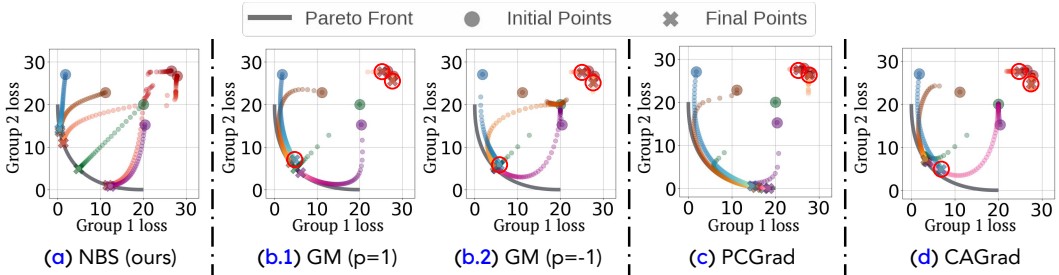

Figure 5: Synthetic illustration of each gradient aggregation method in resolving gradient conflicts and their implication in converging to Pareto front. Red circles imply the nodes that cannot converge to the Pareto front after 1000 steps of updates.

**Why NBS for hypergradient conflict resolution? – An empirical comparison.** As far as we understand, there is no simple way to project or rescale a proposed update (such as LtR, FORML, or Meta-gDRO) to guarantee conflict resolution. Moreover, the NBS finds a balanced solution on the Pareto front where no participant can unilaterally improve their position without others' agreement, which cannot be achieved by a standard way of linear scalarization (e.g., Example 2.27 in [5], and [20]) or other gradient aggregation methods like Generalized Mean (GM) [6]. In synthetic experiments, learning with the NBS effectively steers the model toward the Pareto frontier during early training stages, which is the foundation of our proposed second stage (in our "two-stage" fairness-aware meta-learning) to achieve fairness goals without compromising model performance (the efficacy is illustrated in Figure 3). Empirically comparing optimization trajectories of the NBS and other baselines of gradient aggregation using the same settings in Section 2: GM, PCGrad [55], and CAGrad [29] (Figure 5), we find that these methods, except for the NBS, often favor groups with larger loss magnitudes, resulting in inefficient convergence towards the Pareto frontier. In contrast, the NBS, with its Pareto Optimality axiom, guides all nodes towards efficient convergence by considering the relative importance of different objectives and maximizing joint gains while avoiding favoring large loss value groups at the expense of others' utility.

## A.3   Preliminaries

We give a formal definition of Pareto Optimal and related terms, followed by the details of the four axioms of the NBS.

We write $x \succeq y$ as $x_i \geq y_i$ for all entry $i$ for vector $x, y$, and $x \succ y$ if $x \succeq y$ and $x \neq y$. We use $\| \cdot \|$ to denote $\ell^2$-norm.

**Pareto optimality**. Consider a set of function $f_1, \ldots, f_K$ we want to minimize using some parameter $\theta \in \Theta$. Let vector valued function $f(\theta) = [f_1(\theta) \ldots f_K(\theta)]$. $\theta$ is Pareto optimal if for all other $\theta' \in \Theta$ satisfies $f(\theta') \succeq f(\theta)$. That is, no other $\theta' \in \Theta$ satisfies $f(\theta') \prec f(\theta)$, meaning that objectives cannot be jointly improved without sacrificing any of them.

**Pareto front**. The set of Pareto optimal models forms the Pareto front. There is no preference of the models in the Pareto front, unless with extra customized criteria (such as fairness) introduced.

**Pareto improvement**. $\theta'$ is an Pareto improvement to $\theta$ if $f(\theta') \prec f(\theta)$.

**The four axioms of the NBS:**

1. **Pareto Optimality:** The solution is Pareto optimal in $S$. For solution $x \in S$, it does no exist $y \succ x$ for $y \in S, y \neq x$.

2. **Symmetry:** The solution is invariant to players' order.

3. **Independence of Irrelevant Alternatives:** The solution retains upon superset expansion if it remains in the original set. That is, if we expand the feasible set $S$ and we know the solution stays in $S$, then the solution stays the same.

4. **Invariant to Affine Transformations:** If we apply affine transformations to the utilities, the new solution is the original solution with transformed utilities accordingly. Consider affine transformations $g_1, \ldots, g_K$. The original solution has utility payoffs $[x_1, \ldots, x_K]$. If we transform the utilities from $u_i$ to $g \circ u_i$, the new solution has payoffs $[g_1(x_1), \ldots, g_K(x_K)]$.

### A.4 Problem Setup & Assumptions

**Additional assumptions.** Assume that $S$ is convex and compact, and that there exists an $x \in S$ such that $x_i > d_i$ for all players. We also assume that all players have complete information over the game parameterized as $(S, d)$. Assume without loss of generality $g_i \neq 0$ for $i \in [K]$.

**Definition of $A$.** Recall that in Section 3 we define $A = \{\nabla L : \nabla L \in B_\epsilon, \nabla L_\alpha^\top g_i - D^\top g_i > 0, \forall i \in [K]\}$ with $D = 0$. As a result, $A \subset B_\epsilon$ and $A$'s outer boundary lies on the boundary of $B_\epsilon$. For example, in 2D case, $A$ is a set of circular sectors. A natural question is what happens when considering solutions in $B_\epsilon$. First note that we cannot directly take logarithm to obtain Equation 9 as some $\nabla L_\alpha^\top g_i - D^\top g_i$ are non-positive. Yet, an alternative way is to adjust assumptions on $D$ such that $\nabla L_\alpha^\top g_i - D^\top g_i > 0, i \in [K]$. For example, a way to guarantee the feasibility of $A$ is to set $D = \arg\min_{x \in B_\epsilon} x^\top g_i$ for $i \in [K]$ such that $(\nabla L_\alpha - d)^\top g_i > 0$ for all $i \in [K]$. Here, the feasible region can always be $B_\epsilon$. Assumptions on $D$ will be discussed later in the section. Under current assumption $D = 0$, we can use the axiom of Independence of Irrelevant Alternatives to deduce a result on extending $A$ to $B_\epsilon$: if the new NBS stays in $A$, then it remains unchanged. This would serve as a theoretical guarantee if one wants to make assumptions on the solution under the constraint set $B_\epsilon$ instead.

**Feasibility of $A$.** Another assumption is that $A$ is feasible. If $A$ is empty, then we cannot construct the Bargaining game because it doesn't satisfy Nash's assumption that there always exists one payoff $x$ in the feasible payoff set such that $x$ is better than $d$ for each player. In this case, we use the default fairness protocol $\beta_0$ instead of the improved bargained outcome $\alpha$ as in Algorithm 1. Experiment results show that our mechanism can handle this situation adaptively. In fact, switching to the fairness protocol upon unsuccessful construction of bargaining offers a fresh starting point for subsequent bargaining iterations and helps to escape the saddle points (Figure 6d, also discussed in Section 3.6).

**Assumption $D = 0$.** We assume disagreement payoff $D = 0$. This is to address the hypergradient conflict such that the NBS satisfies $\nabla L_\alpha^\top g_i > 0, i \in [K]$. We now discuss why we don't manually adjust $D$ to make the bargaining problem feasible: Relaxing the constraint set will sacrifice the gradient alignment guarantee and deviate from our focus of conflict resolution. For $D \neq 0$, we are unsure about whether it would help resolve hypergradient conflict. This is because it may flip the sign of some terms, which might result in some $\nabla L_\alpha^\top g_i < 0$ and instead cause hypergradient conflict.

**General $D$.** Though general $D$ may not be suitable for conflict resolution (the problem identified in our paper), it may be useful for other scenarios as a gradient aggregation method (which is out of the scope of this work, but can be a future direction). Here, we include a discussion for completeness: If $A$ is feasible under general $D$, then we are able to get $\sum_i \log (\nabla L_\alpha - D)^\top g_i$ because each term is positive. The derivative w.r.t. $\nabla L_\alpha$ remains $\sum_i \frac{1}{(\nabla L_\alpha - D)^\top g_i} g_i$. Though closed-form solution may not be guaranteed because of the applicability of the tangent slope argument of Equation 5 (that depends on the shape of $A$), but it can be solved as an optimization problem. Note that this may give a very different set coefficient because it is measured in terms of the improvement on $D$.

## A.5 Proofs

**Proof of Theorem** 3.1:

*Theorem.* Under $D = 0$, $\arg\max_{\nabla L_\alpha \in A} \prod_{i \in [K]} (u_i(\nabla L_\alpha) - d_i)$ is achieved at

$$\sum_{i \in [K]} \frac{1}{\nabla L_\alpha^\top g_i} g_i = \gamma \nabla L_\alpha, \quad \text{for some } \gamma > 0. \tag{8}$$

*Proof.* We follow the same steps in Claim 3.1 of [37], along with additional characterization on the shape of $A$ under $D = 0$. Note that this theorem may not work under general $D$.

By the positivity of each term in $A$, this is equivalent to maximizing the summation of logarithms:

$$\arg\max_{\nabla L_\alpha \in B_\epsilon} \sum_{i \in [K]} \log(\nabla L_\alpha^\top g_i). \tag{9}$$

Taking the derivative of the objective w.r.t. $\nabla L_\alpha$ gives $\sum_{i \in K} \frac{1}{\nabla L_\alpha^\top g_i} g_i$. For any $i \in [K]$, we know $x^\top g_i > 0$ if and only if $(cx)^\top g > 0$ with any given $c > 0$. This means that if a point is in $A$, then all points in its radial direction in $B_\epsilon$ are in $A$ (i.e. the boundary of $A$ is a subset of the boundary of $B_\epsilon$). By the Pareto Optimality, the optimal solution must lie on the boundary of $B_\epsilon$ as the utility is monotonically increasing in $\|\nabla L_\alpha\|$ for $\nabla L_\alpha \in A$. The optimal points on the boundary of $B_\epsilon$ have tangent slope 0 (i.e. gradient having the same direction as its normal). Hence, we know the normal is in parallel to $\nabla L_\alpha$ and the desired $\nabla L_\alpha$ satisfies

$$\sum_{i \in [K]} \frac{1}{\nabla L_\alpha^\top g_i} g_i = \gamma \nabla L_\alpha, \quad \text{for some } \gamma > 0. \tag{10}$$

$\square$

**Proof of Theorem** 3.2:

*Theorem.* The solution to Equation 5 is (up to scaling) $\nabla L_\alpha = \sum_{i \in K} \alpha_i g_i$ where

$$G^\top G \alpha = \frac{1}{\alpha} \tag{11}$$

with the element-wise reciprocal $\frac{1}{\alpha}$.

*Proof.* Multiplying both sides with $g_j$, this is equivalent to solving for $x$ in $\sum_{i \in [K]} \frac{g_i^\top g_j}{x^\top g_i} = \gamma x^\top g_j$ for $j \in [K]$. Observe that $x$ is a linear combination of $g_i$ (i.e. $x = \frac{1}{\gamma} \sum_{i \in [K]} (x^\top g_i)^{-1} g_i$). It suffices to solve for coefficients $x^\top g_i$ by the linear system

$$\sum_{i \in K} (g_i^\top g_j)(x^\top g_i)^{-1} = \gamma x^\top g_j \tag{12}$$

for $j \in [K]$. Without loss of generality, set $\gamma = 1$ to ascertain the direction of $x$. Let $\alpha = [\alpha_1 \ldots \alpha_K]$ with $\alpha_i = (x^\top g_i)^{-1}$. Equation 12 becomes

$$g_j^\top \sum_{i \in [K]} g_i \alpha_i = \alpha_j^{-1} \tag{13}$$

for $j \in [K]$, or, equivalently,

$$G^\top G \alpha = \frac{1}{\alpha} \tag{14}$$

with the element-wise reciprocal $\frac{1}{\alpha}$, concluding the proof. Note that $-\alpha$ is also a solution when $\alpha$ is one, yet we preserve $\alpha \in \mathbb{R}_+^K$ (i.e. positive contribution of each $g_i$) in implementation. $\square$

**Additional Note on Linear Independence Condition for Theorem** 3.2:

In Section 3.2, we highlight that linear independence does not hold in general. We also wanted to note that fine-grained subgroup definitions can make linear independence assumptions more realistic as they break down larger groups into smaller, distinct units. If subgroups are well-defined and distinct in their goals, characteristics, or attributes, the vectors representing their goals are more likely to be linearly independent. However, real-world scenarios may still exhibit interdependencies or correlations between subgroup goals, especially in the context of fairness, even if subgroups are finely defined. For example, goals may still rely on common resources or have shared objectives at a higher level. The interdependencies and shared factors may cause dependency in hypergradients. Specifically, if one goal can be expressed as a combination of the goals of other groups, then the corresponding vectors are linearly dependent. Additionally, if the number of subgroups exceeds the dimension of the hypergradient (i.e. definition too fine-grained), it's impossible to have linear independence. Therefore, while fine-grained subgroup definitions can make linear independence assumptions more realistic, it's essential to carefully analyze the specific context and relationships between subgroups to determine the extent to which linear independence holds.

**Proof of Corollary** 3.3:

*Corollary.* (Norm of bargained update) The solution in Theorem 3.2 has $\ell^2$-norm $\sqrt{K}$.

*Proof.* It follows that

$$\|\sum_{i \in K} \alpha_i g_i\|^2 = \sum_{i \in K} \sum_{j \in K} \alpha_i \alpha_j g_i^\top g_j = K. \tag{15}$$

$\square$

**Proof of Corollary** 3.4:

*Corollary.* If $g_j$ is $\sigma$-bounded for $j \in [K]$, $\|\alpha_j^{-1}\|$ is $(\sqrt{K}\sigma)$-bounded for $j \in [K]$.

*Proof.* By Cauchy-Schwarz inequality, we have

$$\|\alpha_j^{-1}\| = \left\| g_j^\top \sum_{i \in [K]} \alpha_i g_i \right\| \leq \left\| g_j^\top \right\| \left\| \sum_{i \in [K]} \alpha_i g_i \right\| \leq \sqrt{K}\sigma. \tag{16}$$

$\square$

**Proof of Theorem** 3.5:

*Theorem.* (Update rule of $\theta$) Denote $L_i^{(train)} = L(D_i^{(t)}|\theta^{(t)}) \in \mathbb{R}$ for the $i$-th sample in training minibatch $D^{(t)}$ at step $t$. $\theta$ is updated as $\theta^{(t+1)} = \theta^{(t)} - \frac{\eta^{(t)}}{|D^{(t)}|} \sum_{i=1}^{|D^{(t)}|} \Delta\theta^i$ with $\Delta\theta^i = \max\left((\nabla_\theta((\beta^{(t)})^\top L^{(val)}))^\top \nabla_\theta L_i^{(train)}, 0\right) \nabla_\theta L_i^{(train)}$.

*Proof.* Since we use the meta-learning framework in [43] with customized $\beta^{(t)}$, we first evaluate the effect of $\beta^{(t)}$ in the computation graph, following similar steps in Equation 12 and Appendix A of [43]. We initialize $\tilde{w}$ to 0. For data sample $i$ in training batch $D^{(t)}$ at step $t$, we perform a single gradient update as in Algorithm 1:

$$\tilde{w}_i^{(t)} = \frac{\partial}{\partial \tilde{w}_i^{(t)}} (\beta^{(t)})^\top L^{(val)}(D^{(val)}|\hat{\theta}^{(t)}) \bigg|_{\tilde{w}_i^{(t)}=0} \tag{17}$$

$$= \frac{\partial}{\partial \theta} (\beta^{(t)})^\top L^{(val)}(D^{(val)}|\theta) \bigg|_{\theta=\theta^{(t)}}^\top \frac{\partial}{\partial w_i^{(t)}} \hat{\theta}^{(t)}(\tilde{w}_i^{(t)}) \bigg|_{\tilde{w}_i^{(t)}=0} \tag{18}$$

$$\propto - \frac{\partial}{\partial \theta} (\beta^{(t)})^\top L^{(val)}(D^{(val)}|\theta) \bigg|_{\theta=\theta^{(t)}}^\top \frac{\partial}{\partial \theta} L(D_i^{(t)}|\theta) \bigg|_{\theta=\theta^{(t)}}. \tag{19}$$

Line 18 comes from chain rule; Line 19 comes from

$$\frac{\partial}{\partial \tilde{w}_i^{(t)}} \hat{\theta}^{(t)}(\tilde{w}_i^{(t)}) = \frac{\partial}{\partial \tilde{w}_i^{(t)}} \left( \theta^{(t)} - \eta^{(t)} \nabla_{\theta^{(t)}} \tilde{w}^{(t)} \cdot L(D^{(t)}|\theta^{(t)}) \right) \tag{20}$$

$$= -\eta^{(t)} \left. \frac{\partial}{\partial \theta} L(D_i^{(t)}|\theta) \right|_{\theta = \theta^{(t)}}. \tag{21}$$

Informally, it means that $\tilde{w}$, or $w$, equivalently, is jointly determined by the $\beta^{(t)}$-weighed gradient on the meta set and the gradient on the training batch. Analogous to Equation 30 of [43], the update rule of $\theta$ is

$$\theta^{(t+1)} = \theta^{(t)} - \eta^{(t)} \frac{\partial}{\partial \theta} \frac{1}{|D^{(t)}|} \sum_{i=1}^{|D^{(t)}|} \max\left(-\tilde{w}_i^{(t)}, 0\right) L(D_i^{(t)}|\theta^{(t)}) \tag{22}$$

$$= \theta^{(t)} - \frac{\eta^{(t)}}{|D^{(t)}|} \sum_{i=1}^{|D^{(t)}|} \Delta\theta^i \tag{23}$$

$$\text{with } \Delta\theta^i = \max\left((\nabla_\theta((\beta^{(t)})^\top L^{(val)}))^\top \nabla_\theta L_i^{(train)}, 0\right) \nabla_\theta L_i^{(train)}. \tag{24}$$

Note that losses and gradients are taken w.r.t. $\theta^{(t)}$, and without loss of generality we can disregard $\eta^{(t)}$ as if it absorbs the normalization constant. $\square$

**Proof of Theorem 3.6:**
*Theorem.* (Pareto improvement of $\tilde{w}$) Use $\alpha^{(t)}$ for the update. Assume $L_i^{(val)}$ is Lipschitz-smooth with constant $C$ and $g_i^{(t)}$ is $\sigma$-bounded at step $t$. If the meta learning rate for $\tilde{w}$ satisfies $\eta^{(t)} \leq \frac{2}{CK\alpha_j^{(t)}}$ for $j \in [K]$, then $L_i^{(val)}(\tilde{w}^{(t+1)}) \leq L_i^{(val)}(\tilde{w}^{(t)})$ for any group $i \in [K]$.

*Proof.* We follow similar (and standard) steps in Theorem 5.4 of [37], yet retrieve a slightly different (and group-wise tighter) upperbound for learning rate. In Theorem 5.4 of [37], the upperbound for learning rate was $\min_{i \in [K]} \frac{1}{CK\alpha_j^{(t)}}$. Our bound is better with multiplicative constant 2 than that of [37].

Write $\Delta\tilde{w} = \tilde{w}^{(t+1)} - \tilde{w}^{(t)}$ and use a well-known property of Lipschitz-smoothness:

$$L_i^{(val)}(\tilde{w}^{(t+1)}) \leq L_i^{(val)}(\tilde{w}^{(t)}) - I_1 + I_2 \tag{25}$$

$$\text{with } I_1 = \eta^{(t)} \nabla L_i^{(val)}(\tilde{w}^{(t)})^\top \Delta\tilde{w} \tag{26}$$

$$= \eta^{(t)} g_i^{(t)\top} \sum_{j=1}^K \alpha_j^{(t)} g_j^{(t)} \tag{27}$$

$$= \frac{\eta^{(t)}}{\alpha_i^{(t)}}, \tag{28}$$

$$I_2 = \frac{C}{2} \|\eta^{(t)} \Delta\tilde{w}\|^2. \tag{29}$$

By Corollary 3.3,

$$I_2 = \frac{C}{2} \|\eta^{(t)} \Delta\tilde{w}\|^2 = \frac{C(\eta^{(t)})^2}{2} \|\Delta\tilde{w}\|^2 = \frac{CK(\eta^{(t)})^2}{2}. \tag{30}$$

Observe that $-I_1 + I_2 \leq 0$ is equivalent to

$$\frac{CK(\eta^{(t)})^2}{2} \leq \frac{\eta^{(t)}}{\alpha_i^{(t)}} \tag{31}$$

$$\eta^{(t)} \leq \frac{2}{CK\alpha_i^{(t)}}. \tag{32}$$

We also have $\frac{2}{CK\alpha_i^{(t)}} \leq \frac{2\sqrt{K}\sigma}{CK} = \frac{2\sigma}{C\sqrt{K}}$ by Corollary 3.4. Note that $L_i^{(val)}(\tilde{w}^{(t+1)}) \leq L_i^{(val)}(\tilde{w}^{(t)})$ is strict when Inequality 32 is strict, which yields Pareto Improvement. We just perform one step of gradient update of $\tilde{w}$ in Algorithm 1, yet this result is applicable to single level optimization that directly optimize the parameter in interest. □

**Proof of Theorem 3.7**: We will use the following corollary in the proof:

**Corollary A.1.** *Assume $f : \mathbb{R}^d \to \mathbb{R}^K$ is Lipschitz-smooth with constant $C$. Fix $\beta \in \mathbb{R}^K$ with finite $\|\beta\|$. For $g : \mathbb{R}^K \to \mathbb{R}$, $g(x) = \beta^\top f(x)$. Then $g$ is Lipschitz-smooth with constant $C\|\beta\|$.*

*Proof.* This is a classical result in Real Analysis. By chain rule, we have $\nabla g(x) = (\nabla f(x))\beta$. Then for any $x, y \in \mathbb{R}^d$, by Cauchy-Schwarz inequality,

$$\|\nabla g(x) - \nabla g(y)\| = \|(\nabla f(x))\beta - (\nabla f(y))\beta\| = \|(\nabla f(x) - \nabla f(y))\beta\| \leq \|\nabla f(x) - \nabla f(y)\|\|\beta\| \leq C\|\beta\|\|x - y\|. \tag{33}$$

Thus $g$ is Lipschitz-smooth with constant $C\|\beta\|$. □

*Theorem.* (Monotonic improvement of validation loss w.r.t. $\theta$) Assume $L^{(val)}$ is Lipschitz-smooth with constant $C$ and $\nabla_\theta L_i^{(train)}$ is $\sigma$-bounded. If the learning rate for $\theta$ satisfies $\eta^{(t)} \leq \frac{2|D^{(t)}|}{C\|\beta^{(t)}\|\sigma^2}$, then $L_{\beta^{(t)}}(\theta^{(t+1)}) \leq L_{\beta^{(t)}}(\theta^{(t)})$ for any fixed vector $\beta^{(t)}$ with finite $\|\beta^{(t)}\|$ used to update $\theta^{(t)}$.

*Proof.* We incorporate $\beta^{(t)}$ in the computation when following the same (and standard) steps in Lemma 1 of [43]. We drop the superscript of $\beta^{(t)}$ for simplicity. By Corollary A.1, we know $L_\beta$ is Lipschitz-smooth with constant $C\|\beta\|$. Similar to Theorem 3.6, we use the gradients derived in Theorem 3.5 and have

$$L_\beta(\theta^{(t+1)}) \leq L_\beta(\theta^{(t)}) - I_1 + I_2 \tag{34}$$

$$\text{with } I_1 = (\nabla L_\beta)^\top \Delta\theta \tag{35}$$

$$= (\nabla L_\beta)^\top \frac{\eta^{(t)}}{|D^{(t)}|} \sum_{i \in D^{(t)}} \max\{(\nabla L_\beta)^\top \nabla L_i^{(train)}, 0\} \nabla L_i^{(train)} \tag{36}$$

$$= \frac{\eta^{(t)}}{|D^{(t)}|} \sum_{i \in D^{(t)}} \max\{((\nabla L_\beta)^\top \nabla L_i^{(train)})^2, 0\}, \tag{37}$$

$$I_2 = \frac{C\|\beta\|}{2} \|\Delta\theta\|^2 \tag{38}$$

$$= \frac{C\|\beta\|}{2} \left\| \frac{\eta^{(t)}}{|D^{(t)}|} \sum_{i \in D^{(t)}} \max\{(\nabla L_\beta)^\top \nabla L_i^{(train)}, 0\} \nabla L_i^{(train)} \right\|^2 \tag{39}$$

$$\leq \frac{C\|\beta\|}{2} \frac{(\eta^{(t)})^2}{|D^{(t)}|^2} \sum_{i \in D^{(t)}} \left\| \max\{(\nabla L_\beta)^\top \nabla L_i^{(train)}, 0\} \nabla L_i^{(train)} \right\|^2 \tag{40}$$

$$\leq \frac{C\|\beta\|}{2} \frac{(\eta^{(t)})^2}{|D^{(t)}|^2} \sum_{i \in D^{(t)}} \max\{((\nabla L_\beta)^\top \nabla L_i^{(train)})^2, 0\} \left\| \nabla L_i^{(train)} \right\|^2 \tag{41}$$

$$\leq \frac{C\|\beta\|(\eta^{(t)})^2\sigma^2}{2|D^{(t)}|^2} \sum_{i \in D^{(t)}} \max\{((\nabla L_\beta)^\top \nabla L_i^{(train)})^2, 0\}. \tag{42}$$

Line 40 and 41 come from the triangle inequality and Cauchy-Schwarz inequality. Note that here the gradients are taken w.r.t. $\theta$. We know $-I_1 + I_2 \leq 0$ is equivalent to

$$\eta^{(t)} \leq \frac{2|D^{(t)}|}{C\|\beta\|\sigma^2}, \tag{43}$$

concluding the proof. Here, we generalize Lemma 1 of [43] from a fixed protocol to any protocol $\beta$ with finite norm. Setting the bargained $\alpha$ to $\beta$ gives a new desirable property of the NBS in

mete-learning, which establishes its validity as a meta-learning protocol. As such, it is a novel result of the NBS on the monotonicity of validation loss, which has not been presented before as [37] did not focus on meta-learning. □

### A.6 Detailed Experimental Settings

**Synthetic settings (used in §2, §4).** We provide here the details for the illustrative example of Figure 2b and 3. We use a modified version of the illustrative example in [37]. We first present our modified learning problem: Let $\theta = (\theta_1, \theta_2) \in \mathbb{R}^2$, and consider the following objectives:

$$\ell_1(\theta) = c_1(\theta)f_1(\theta) + c_2(\theta)g_1(\theta) \text{ and } \ell_2(\theta) = c_1(\theta)f_2(\theta) + c_2(\theta)g_2(\theta), \text{ where}$$
$$f_1(\theta) = \log(\max(|0.5(-\theta_1 - 7) - \tanh(-\theta_2)|, 5e - 6)) + 6,$$
$$f_2(\theta) = \log(\max(|0.5(-\theta_1 + 3) - \tanh(-\theta_2) + 2|, 5e - 6)) + 6,$$
$$g_1(\theta) = ((-\theta_1 + 7)^2 + 0.1 \cdot (-\theta_2 - 8)^2)/10 - 20,$$
$$g_2(\theta) = ((-\theta_1 - 7)^2 + 0.1 \cdot (-\theta_2 - 8)^2)/10 - 20,$$
$$c_1(\theta) = \max(\tanh(0.5\theta_2), 0) \text{ and } c_2(\theta) = \max(\tanh(-0.5\theta_2), 0)$$

We now set $L_1 = \ell_1$ and $L_2 = \ell_2$ as our objectives to simulate the case where two participated groups have the same scale of loss, as a closer reflection of the loss scope may encountered during inter-group bargaining. In particular, with the loss scale being set to the same, the fairness goal of the evaluated notions, i.e., LtR, FORML, and Meta-gDRO, becomes the same as **x=y**. We use six different initialization points $\{(-8.5, 7.5), (0.0, 0.0), (9.0, 9.0), (-7.5, -0.5), (9, -1.0), (9, -20)\}$. We use the SGD optimizer and train each method for 1000 iterations with a learning rate of 0.1.

**Standard fairness benchmark settings (used in §4).** The detailed fairness dataset settings are provided in Table 2. We access these fairness datasets via Library for Semi-Automated Data Science: https://github.com/IBM/lale. The library is under Apache 2.0 License.

| Dataset | # samples | # features | Favor label | Attribute | Training Set | Test set (3% of whole data) | Validation set |
|---|---|---|---|---|---|---|---|
| **Adult Income** | 48842 | 105 | 1 | sex | M ({0: 22365, 1: 9555}) 
 F ({0: 14060, 1: 1430}) | M ({0: 366, 1: 366}) 
 F ({0: 366, 1: 366}) | M ({0: 3, 1: 3}) 
 F ({0: 3, 1: 3}) |
| | | | | race | Black ({0: 3972, 1: 428}) 
 White ({0: 31006, 1: 10458}) 
 Asian ({0: 969, 1: 282}) 
 Other ({0: 233}) 
 Amer-Indian ({0: 289, 1: 1 }) | Black ({0: 146, 1: 146}) 
 White ({0: 146, 1: 146}) 
 Asian ({0: 146, 1: 146}) 
 Other ({0: 146, 1: 146}) 
 Amer-Indian ({0: 146, 1: 146}) | Black ({0: 3, 1: 3}) 
 White ({0: 3, 1: 3}) 
 Asian ({0: 3, 1: 3}) 
 Other ({0: 3, 1: 3}) 
 Amer-Indian ({0: 3, 1: 3}) |
| **Bank Telemarketing** | 45211 | 51 | 1 | age | age > 25 ({1: 38564, 0: 4632}) 
 age < 25 ({1: 715, 0: 108}) | age > 25 ({1: 339, 0: 339}) 
 age < 25 ({1: 339, 0: 339}) | age > 25 ({1: 3, 0: 3}) 
 age < 25 ({1: 3, 0: 3}) |
| **Credit Default** | 30000 | 24 | 0 | sex | M ({1: 14122, 0: 3542}) 
 F ({1: 8787, 0: 2656}) | M ({1: 225, 0: 225}) 
 F ({1: 225, 0: 225}) | M ({1: 4, 0: 4}) 
 F ({1: 4, 0: 4}) |
| **Communities and Crime** | 1994 | 1929 | 0 | blackgt6pct | False ({1: 1002, 0: 0 }) 
 True ({1: 841, 0: 101}) | False ({1: 14, 0: 14}) 
 True ({1: 14, 0: 14}) | False ({1: 1, 0: 1}) 
 True ({1: 1, 0: 1}) |
| **Titanic Survival** | 1309 | 1526 | 1 | sex | M ({1: 151, 0: 672}) 
 F ({1: 329, 0: 118}) | M ({1: 9, 0: 9}) 
 F ({1: 9, 0: 9}) | M ({1: 1, 0: 1}) 
 F ({1: 1, 0: 1}) |
| **Student Performance** | 649 | 58 | 1 | sex | M ({1: 200, 0: 33}) 
 F ({1: 315, 0: 32}) | M ({1: 16, 0: 16}) 
 F ({1: 16, 0: 16}) | M ({1: 2, 0: 2}) 
 F ({1: 2, 0: 2}) |

Table 2: We detail the number of samples per group, categorized by protected attributes and labels. The student performance dataset allocates 10% for testing due to its smaller size, while others reserve 3%. Notably, in the adult income training dataset with race as a protected attribute, only one Amer-Indian sample with the positive (favorable) label. Additionally, after balancing test and validation sets, the community and crime dataset has no samples in the "False" group labeled "0". These training data distribution-wise problems are marked in yellow and were maintained to examine how extreme imbalances in training data impact various algorithms.

**Training specifics for standard fairness benchmarks (used in §4).** For the models applied to these fairness datasets, we consistently employ a 3-layer neural network architecture comprised of an input layer, one hidden layer, and an output layer. The input layer takes in features and transforms them to a dimension of size 128. A ReLU activation function and a dropout layer for regularization follow this. The hidden layer further processes the data, again followed by a ReLU and dropout layer. Finally, the output layer maps the representation from the hidden layer to the number of classes

specified (2 for the considered fairness datasets). This architecture utilizes dropout after each ReLU activation to reduce overfitting, making it suitable for classification tasks.

We fine-tuned the training hyperparameters based on the baseline model. Common hyperparameters across all algorithms include a total of 50 training epochs, an SGD optimizer momentum of 0.9, and a weight decay of 5e-4, with the bargaining phase limited to 15 epochs for the three settings incorporating proposed Nash-Meta-Learning. Hyperparameters that varied are detailed in Table 3.

**Switching phases and selection of** $T_{bar}$**.** In practice, we determine $T_{bar}$ by monitoring the bargaining success rate. We observed from real-data experiments that model's performance would stabilize when this rate stabilizes. This may serve as a sign of switching to Stage 2 because no significant improvements could be brought by bargaining. In our real-data experiments, we set $T_{bar}$ to 15 epochs, as this allowed all evaluated settings to reach a stable bargaining success rate.

**Computing.** All experiments were conducted on an internal cluster using one chip of H-100.

| Dataset | Attribute | Learning rate | Dropout Probability | Batch Size |
|---|---|---|---|---|
| Adult Income | sex | 1e-3 | 0.2 | 512 |
| Adult Income | race | 5e-4 | 0.4 | 512 |
| Bank Telemarketing | age | 1e-3 | 0.3 | 512 |
| Credit Default | sex | 1e-3 | 0.5 | 512 |
| Communities and Crime | blackgt6pact | 1e-4 | 0.2 | 32 |
| Titanic Survival | sex | 1e-3 | 0.4 | 32 |
| Student Performance | sex | 1e-3 | 0.05 | 32 |

Table 3: Hyperparameter settings for standard fairness datasets.

## A.7 Additional Results

### A.7.1 Results with 95% Confidence Intervals (CIs).

The main empirical results are averaged from five independent runs using different random seeds, the 95% CI marked in Table 4. In addition to the key findings shown in Table 1, we find the results from the two-stage method are often more stable than the one-stage baselines, reflected by tighter CIs.

### A.7.2 Additional Analysis

**Noise analysis.** To analyze the noise presented in validation sets, as referenced in Table 5, we employ normalized mutual information [24]. This metric offers a measure of the shared information between features and labels, normalized by the sum of their individual entropies. The normalized mutual information $I_{norm}$ between a feature set $X$ and labels $Y$ is computed as:

$$I_{norm}(X;Y) = \frac{I(X;Y)}{H(X) + H(Y)}$$

where $I(X;Y)$ is the mutual information between $X$ and $Y$, representing the amount of shared information, and $H(X)$ and $H(Y)$ are the entropies of the feature set and the labels, rsp. A higher $I_{norm}$ indicates less noise, signifying a stronger relationship between the features and labels. We referred to https://github.com/mutualinfo/mutual_info (Apache-2.0 license) for implementation.

Referring to Table 5, it is evident that the validation set used from Credit Default dataset and the Communities and Crime dataset exhibits a notably lower mutual information score, compared to the other datasets. This suggests a significantly weaker correlation between the features and labels in these particular validation sets and a higher level of noise. Such a disparity in mutual information underscores the challenge in achieving high model performance and could explain the varying degrees of success observed across different fairness interventions within this dataset.

| | | Baseline | DRO | LtR | | FORML | | Meta-gDRO | |
|---|---|---|---|---|---|---|---|---|---|
| | | | | one-stage | two-stage (ours) | one-stage | two-stage (ours) | one-stage | two-stage (ours) |
| **I. Adult Income** [3], (sensitive attribute: **sex**) | | | | | | | | | |
| Overall AUC (↑) | | $0.778 \pm 0.004$ | $0.761 \pm 0.010$ | $0.830 \pm 0.002$ | $0.830 \pm 0.004$ | $0.801 \pm 0.009$ | $0.810 \pm 0.004$ | $0.810 \pm 0.004$ | $0.837 \pm 0.004$ |
| Max-gAUCD (↓) | | $0.016 \pm 0.012$ | $0.029 \pm 0.016$ | $0.050 \pm 0.005$ | $0.047 \pm 0.003$ | $0.075 \pm 0.042$ | $0.031 \pm 0.009$ | $0.052 \pm 0.007$ | $0.046 \pm 0.004$ |
| Worst-gAUC (↑) | | $0.770 \pm 0.011$ | $0.747 \pm 0.018$ | $0.805 \pm 0.004$ | $0.807 \pm 0.004$ | $0.763 \pm 0.022$ | $0.795 \pm 0.007$ | $0.809 \pm 0.006$ | $0.814 \pm 0.006$ |
| **II. Adult Income** [3], (sensitive attribute: **race**; * one group in training data contains only one positive (favorable) label. ) | | | | | | | | | |
| Overall AUC (↑) | | $0.668 \pm 0.004$ | $0.652 \pm 0.005$ | $0.803 \pm 0.005$ | $0.805 \pm 0.007$ | $0.710 \pm 0.023$ | $0.775 \pm 0.008$ | $0.775 \pm 0.013$ | $0.793 \pm 0.004$ |
| Max-gAUCD (↓) | | $0.225 \pm 0.027$ | $0.236 \pm 0.011$ | $0.090 \pm 0.015$ | $0.090 \pm 0.018$ | $0.290 \pm 0.006$ | $0.134 \pm 0.040$ | $0.163 \pm 0.044$ | $0.158 \pm 0.029$ |
| Worst-gAUC (↑) | | $0.544 \pm 0.019$ | $0.538 \pm 0.002$ | $0.755 \pm 0.004$ | $0.760 \pm 0.018$ | $0.540 \pm 0.019$ | $0.688 \pm 0.036$ | $0.694 \pm 0.047$ | $0.703 \pm 0.028$ |
| **III. Bank Telemarketing** [33], (sensitive attribute: **age**) | | | | | | | | | |
| Overall AUC (↑) | | $0.697 \pm 0.004$ | $0.686 \pm 0.015$ | $0.724 \pm 0.009$ | $0.728 \pm 0.005$ | $0.706 \pm 0.101$ | $0.779 \pm 0.027$ | $0.698 \pm 0.006$ | $0.722 \pm 0.009$ |
| Max-gAUCD (↓) | | $0.013 \pm 0.010$ | $0.025 \pm 0.010$ | $0.099 \pm 0.012$ | $0.083 \pm 0.009$ | $0.039 \pm 0.023$ | $0.029 \pm 0.009$ | $0.098 \pm 0.010$ | $0.079 \pm 0.008$ |
| Worst-gAUC (↑) | | $0.691 \pm 0.008$ | $0.691 \pm 0.008$ | $0.675 \pm 0.013$ | $0.686 \pm 0.005$ | $0.686 \pm 0.090$ | $0.764 \pm 0.025$ | $0.649 \pm 0.009$ | $0.683 \pm 0.010$ |
| **IV. Credit Default** [54], (sensitive attribute: **sex**; * val data is more noisy than others, see Table 5, Appendix A.7. ) | | | | | | | | | |
| Overall AUC (↑) | | $0.634 \pm 0.004$ | $0.624 \pm 0.008$ | $0.630 \pm 0.019$ | $0.616 \pm 0.035$ | $0.554 \pm 0.039$ | $0.611 \pm 0.018$ | $0.682 \pm 0.003$ | $0.661 \pm 0.017$ |
| Max-gAUCD (↓) | | $0.024 \pm 0.011$ | $0.022 \pm 0.010$ | $0.037 \pm 0.022$ | $0.025 \pm 0.012$ | $0.017 \pm 0.012$ | $0.033 \pm 0.018$ | $0.016 \pm 0.007$ | $0.022 \pm 0.016$ |
| Worst-gAUC (↑) | | $0.622 \pm 0.005$ | $0.613 \pm 0.012$ | $0.612 \pm 0.013$ | $0.603 \pm 0.032$ | $0.545 \pm 0.033$ | $0.595 \pm 0.010$ | $0.674 \pm 0.004$ | $0.650 \pm 0.025$ |
| **V. Communities and Crime** [42], (sensitive attribute: **blackgt6pct**; * val data is noisy, meanwhile one group in training data are all positive labels ) | | | | | | | | | |
| Overall AUC (↑) | | $0.525 \pm 0.008$ | $0.568 \pm 0.006$ | $0.679 \pm 0.022$ | $0.700 \pm 0.032$ | $0.554 \pm 0.010$ | $0.568 \pm 0.018$ | $0.686 \pm 0.035$ | $0.686 \pm 0.035$ |
| Max-gAUCD (↓) | | $0.050 \pm 0.015$ | $0.136 \pm 0.012$ | $0.071 \pm 0.045$ | $0.129 \pm 0.073$ | $0.107 \pm 0.020$ | $0.136 \pm 0.037$ | $0.114 \pm 0.073$ | $0.114 \pm 0.073$ |
| Worst-gAUC (↑) | | $0.500 \pm 0.000$ | $0.500 \pm 0.000$ | $0.643 \pm 0.000$ | $0.636 \pm 0.046$ | $0.500 \pm 0.000$ | $0.500 \pm 0.000$ | $0.629 \pm 0.042$ | $0.629 \pm 0.042$ |
| **VI. Titanic Survival** [12], (sensitive attribute: **sex**) | | | | | | | | | |
| Overall AUC (↑) | | $0.972 \pm 0.000$ | $0.983 \pm 0.012$ | $0.967 \pm 0.010$ | $0.978 \pm 0.010$ | $0.950 \pm 0.018$ | $0.972 \pm 0.022$ | $0.961 \pm 0.012$ | $0.972 \pm 0.016$ |
| Max-gAUCD (↓) | | $0.056 \pm 0.000$ | $0.033 \pm 0.024$ | $0.044 \pm 0.019$ | $0.044 \pm 0.019$ | $0.033 \pm 0.024$ | $0.011 \pm 0.019$ | $0.033 \pm 0.024$ | $0.033 \pm 0.024$ |
| Worst-gAUC (↑) | | $0.944 \pm 0.000$ | $0.967 \pm 0.024$ | $0.944 \pm 0.000$ | $0.956 \pm 0.019$ | $0.933 \pm 0.019$ | $0.967 \pm 0.024$ | $0.944 \pm 0.000$ | $0.956 \pm 0.019$ |
| **VII. Student Performance** [11], (sensitive attribute: **sex**) | | | | | | | | | |
| Overall AUC (↑) | | $0.784 \pm 0.011$ | $0.816 \pm 0.020$ | $0.900 \pm 0.007$ | $0.900 \pm 0.011$ | $0.828 \pm 0.026$ | $0.822 \pm 0.032$ | $0.909 \pm 0.020$ | $0.912 \pm 0.017$ |
| Max-gAUCD (↓) | | $0.119 \pm 0.032$ | $0.106 \pm 0.037$ | $0.013 \pm 0.013$ | $0.037 \pm 0.027$ | $0.056 \pm 0.032$ | $0.031 \pm 0.025$ | $0.031 \pm 0.018$ | $0.025 \pm 0.011$ |
| Worst-gAUC (↑) | | $0.725 \pm 0.020$ | $0.762 \pm 0.032$ | $0.894 \pm 0.013$ | $0.881 \pm 0.020$ | $0.800 \pm 0.022$ | $0.806 \pm 0.020$ | $0.894 \pm 0.028$ | $0.900 \pm 0.020$ |

Table 4: Performance comparison on standard fairness datasets (averaged from 5 runs), with 95% CI.

**Additional illustration of resolving hypergradient conflict and our dynamics.** Figure 6 provides additional visualizations of the impact of our bargaining strategy on gradient conflict resolution, utilizing the adult income dataset with sex as the sensitive attribute. The data indicates that FORML and Meta-gDRO exhibit greater initial conflicts compared to LtR, but also receive more pronounced benefits from our bargaining steps. In Figure 6d, we observe instances where bargaining was unsuccessful, yet an external force, aligned with specific fairness goals (shown with Meta-gDRO as an example), aids in overcoming local sticking points. These observations not only substantiate the utility of bargaining steps in mitigating conflicts but also empirically demonstrate the dynamic process of opting for updates over stagnation, which aligns with the analysis of our method discussed in §3.6.

| | Attribute | # features | Normalized Mutual Information (Validation Set) |
|---|---|---|---|
| **Adult Income** | sex | 105 | 31.007 e-5 |
| | race | 105 | 43.337 e-5 |
| **Bank Telemarketing** | age | 51 | 679.576 e-5 |
| **Credit Default** | sex | 24 | **26.023 e-5** |
| **Communities and Crime** | blackgt6pact | 1929 | **9.962 e-5** |
| **Titanic Survival** | sex | 1526 | 63.911 e-5 |
| **Student Performance** | sex | 58 | 253.198 e-5 |

Table 5: Mutual information of each dataset's validation set normalized by the summation of features' and labels' entropy.

**Sensitivity to the initial weighting protocol $\beta_0$.** Figure 4 demonstrates that the improvement from bargaining correlates with the initial hypergradient alignment rate (the portion of aligned batches).

|  | LtR | FORML | Meta-gDRO |
|---|---|---|---|
| Hypergradients Aligned | 759 (59.5%) | 0 (0%) | 365 (28.6%) |
| $A$ Non-Empty | 1046 (82.0%) | 974 (76.4%) | 1072 (84.1%) |

Table 6: One-stage methods in the first 15 epochs on US Crime dataset. "Hypergradients aligned" means the proposed update of the batch lies in $A$ ($\nabla L_\beta^\top g_i > 0$ for all $i$). Percentage in parethesis is the portion out of total number of batches (1275 batches). We observed that approximately 80% of time the $A$ is nonempty, which gives room for improvement with bargaining.

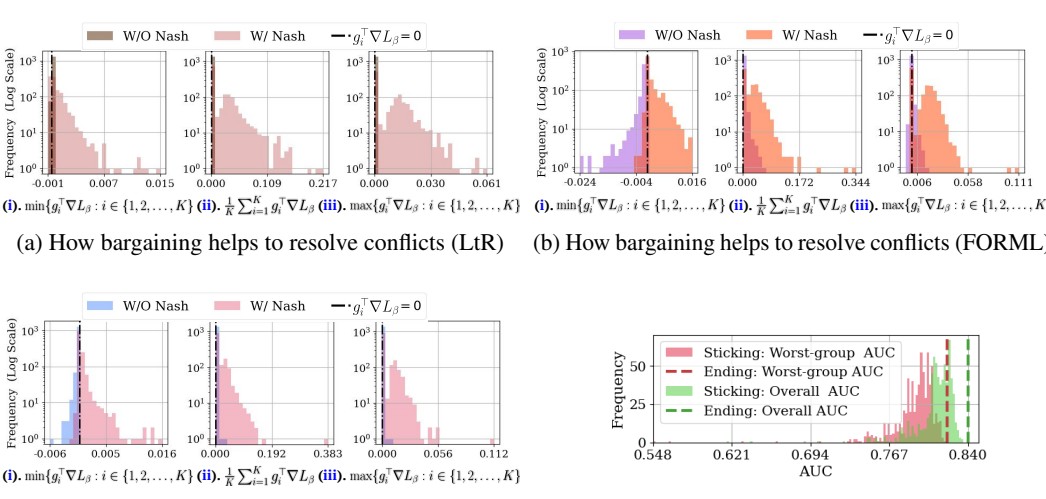

(a) How bargaining helps to resolve conflicts (LtR)   (b) How bargaining helps to resolve conflicts (FORML)

(c) How bargaining helps to resolve conflicts (Meta-gDRO)

(d) Bargaining failed (local sticking points) and how external force helps (Meta-gDRO with Nash)

Figure 6: Visualizations of how Nash bargaining improves gradient conflicts across different methods (adult income with sex as the sensitive attribute). With (i), (ii), (iii) in (a), (b), (c) depicting the least value of group-wise hypergradient alignment ($\min\{g_i^\top \nabla L_\beta : i \in [K]$), the averaged group-wise hypergradient alignment value ($\frac{1}{K} \sum_{i \in [K]} g_i^\top \nabla L_\beta$), and the maximum value of group-wise hypergradient alignment ($\max\{g_i^\top \nabla L_\beta : i \in [K]\}$) respectively.

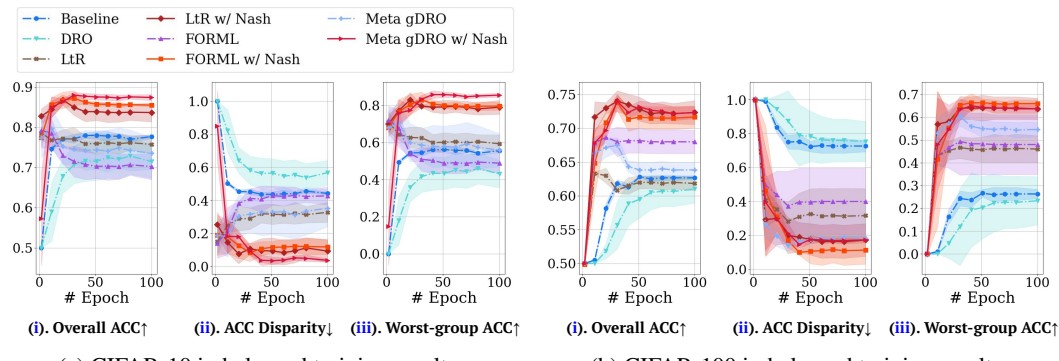

(a) CIFAR-10 imbalanced training results.   (b) CIFAR-100 imbalanced training results.

Figure 7: Comparative analysis of accuracy and fairness during CIFAR-10 and CIFAR-100 imbalanced training (results averaged from 5 seeds). Subfigures (a) depict the CIFAR-10 results and (b) show CIFAR-100. Each plot tracks the evolution of overall ACC, ACC disparity, and worst-group ACC over training epochs, highlighting performance dynamics under a 99%-1% class imbalance for CIFAR-10 and a 95%-5% for CIFAR-100. The shaded regions represent the 95% CI, offering insight into the consistency of each method. The efficacy of Nash bargaining in the two-stage approach is evident, particularly in enhancing the worst-group ACC, vital for fairness in imbalanced settings.

When this initial rate is low, the bargaining process yields significant improvements (for example, FORML). Conversely, when the initial alignment rate is high, the gains from bargaining are more

modest. This relationship may provide insight into the varying effectiveness of our approach across different scenarios.

**Computational costs.** Our analysis shows that the entire training process takes approximately 1.2-1.4 times as long as training with bargaining does, with the bargaining steps takes 2-10 times as regular training takes per epoch. We view this as a worthwhile trade-off given the enhanced performance and fairness achieved.

### A.7.3 Imbalanced Image Classification

To further validate our framework, we extend our evaluation to imbalanced image classification on subsets of the CIFAR-10 and CIFAR-100 datasets [25]. For CIFAR-10, we construct a training set of 5000 samples with an imbalanced class distribution of 99% to 1%. Due to the smaller per-class sample size in CIFAR-100, we create a 500-sample training set with a 95%-5% split. Test sets are balanced, containing an equal number of samples from each class (1000 for CIFAR-10, 100 for CIFAR-100), and validation sets are composed of five samples from each group. We utilize ResNet-18 [19] for modeling and employ SGD with a learning rate of 5e-4, momentum of 0.9, and a weight decay of 5e-4. All methods are trained for 100 epochs with a batch size of 128, and for two-stage algorithms, the initial 30 epochs incorporate bargaining.

Figure 7a delineates the trajectory of accuracy (ACC) for different methods during training epochs under a 99%-1% class imbalance on CIFAR-10. One-stage meta-learning methods encounter difficulties in converging to a performing and fair model, often underperforming in terms of both ACC and fairness in this context. In contrast, our two-stage method not only achieves significant improvements in ACC and fairness but also enhances algorithmic stability, as indicated by the tighter CIs. The early application of bargaining secures a higher initial ACC, and this advantage is sustained even as the model later transitions to focus on specific fairness objectives. The consistently improved trajectory, echoing our synthetic example insights (Figure 3), suggests that reaching the Pareto front and subsequently focusing on fairness does not degrade the model's performance. The above observations underscore the efficacy of the two-stage bargaining approach in managing the trade-offs inherent in fairness-focused tasks. The challenge of achieving fairness across groups is amplified by the granular nature of CIFAR-100 (more classes and fewer samples per class) (Figure 7b). Despite this, the two-stage Nash bargaining approach demonstrates a substantial improvement in aligning model performance with fairness goals. In contrast, the one-stage baseline methods show limitations in addressing the imbalance, as evidenced by their fluctuating and often lower ACC trajectories. Nash bargaining yields a marked improvement in the worst-group ACC, which is critical in such imbalanced scenarios. This indicates that the bargaining phase helps the model to better navigate the complex decision space of CIFAR-100, providing a more equitable distribution of predictive accuracy among the classes. Notably, the early-stage bargaining not only propels the model towards higher initial accuracy but also prepares it to maintain performance when the fairness objectives become the primary focus in the later stages of training. This strategic approach ensures that the pursuit of fairness does not come at the expense of overall accuracy, a balance that is particularly challenging in the diverse and imbalanced CIFAR-100 environment. The above results from CIFAR-100 reinforce the generalizable effectiveness of Nash bargaining in our proposed two-stage meta-learning.

### A.8 Broader Impact

In this study, we present advancements in ML fairness through the development and analysis of new fairness-aware meta-learning methods. Our research strictly utilizes synthetic or publicly accessible open-source datasets, avoiding the use of human subjects. While our method demonstrates notable improvements over existing approaches in various experimental setups, we recognize that it is not infallible. As such, any application of our method in real-world decision-making tasks must be approached with meticulous consideration of the specific context and potential repercussions. It's crucial to understand that achieving ML fairness is an ongoing, collective endeavor within the research community.

Moreover, we also acknowledge that the fairness-aware meta-learning methods developed in this study are not panaceas for all fairness-related issues in machine learning. There are still several challenges and open research directions in this area, such as addressing fairness in multimodal learning, developing better explainability and interpretability techniques, and ensuring fairness

in model adaptation and transfer learning. Furthermore, there is a need for more diverse and representative datasets to evaluate the effectiveness and generalizability of fairness-aware methods. To address these challenges, we call for further research and collaboration among researchers, practitioners, and policymakers to advance the state-of-the-art in ML fairness and to ensure that AI systems are fair, transparent, and accountable for all individuals and groups.

