# OpenReview forum: "Fairness-Aware Meta-Learning via Nash Bargaining"
_NeurIPS.cc/2024/Conference — NeurIPS 2024 poster_

### Official Review · Reviewer_RvEe · 2024-07-11

**Soundness:** 3
**Presentation:** 4
**Contribution:** 3
**Rating:** 6
**Confidence:** 3

**Summary:**

This paper tried to address the problem of hypergradient conflicts in fairness-aware meta-learning, where the overall validation loss gradient is not aligned with per-group validation loss. They do this by using Nash Bargaining to allow the different groups to achieve consensus on the gradient update direction. Their approach is to have a two-stage solution, where initial rounds of optimization drive the solution to a pareto front, following which they optimize for fairness. They show useful properties of this approach, including non-reliance on the linear independence assumption previously used in NBS-based gradient aggregation.  Finally, they show empirical results in multiple datasets by adding their two-stage approach to existing algorithms, showing improvement in multiple domains.

**Strengths:**

The paper is well-written. The visualizations for their experiments are able to convey a lot of information in a small figure, and decently complement the claims of the paper, allowing readers to intuitively understand what the approach is doing. They derive a closed-form update rule for the weights resulting from the Nash bargaining. The problem being tackled is also significant, adding weight to the contributions of the paper.

**Weaknesses:**

1. It is not very clear to me when the algorithm should move onto the second stage. How is the threshold number of steps determined before moving from the Nash bargaining to optimizing fairness? Or are both steps always performed? Some text makes it seem like they happen independently, but this is not clear from the description of the method, or through the algorithm.
2. The histograms (e.g. Figure 1, right) are unclear. Are the two colors ever overlapping? If they are disjoint, why is that the case? If they do overlap, there should be some indication of the hidden bars.

**Questions:**

1. Please address the questions from the weaknesses section.
2. Since the closed-form update is identical to previous work, is the only novelty the removal of the linear independence assumption? Do previous methods' results also match what this paper's experimental section shows?

**Limitations:**

Yes

---

> ### Author Rebuttal · Authors · 2024-08-04
>
> Thank you for the reviewer’s constructive feedback and acknowledgment of our contribution. We appreciate the opportunity to clarify our approach and address your concerns.
>
> ____
> **W1: Transitioning to Stage 2 and selection of $T_{bar}$.**
>
> Thank you for your question! In practice, we determine $T_{bar}$ by monitoring the bargaining success rate. We observed from real-data experiments that the model's performance would stabilize when this rate stabilizes. This may serve as a sign of switching to Stage 2 because no significant improvements could be brought by bargaining. In our real-data experiments, we set $T_{bar}$ to 15 epochs, as this allowed all evaluated settings to reach a stable bargaining success rate. We also want to clarify that $\beta_0$ is also used in Stage 1 when bargaining fails, which would preserve the fairness objectives meanwhile providing a fresh start for subsequent bargaining. We've updated the manuscript to improve clarity and consistency.
>
> ___
> **W2: Figure 1.b clarification.**
>
> The two colors are overlapping. We've updated the figure in our manuscript.
>
> ___
> **Q2.1: Theoretical novelty.**
>
> Thank you for your question! The removal of linear independence in Thm 3.2 is *not* the only novelty. Compared to previous work [37], our additional theoretical contribution is as follows:
>
>  - For Pareto improvement of $\tilde{w}$, Thm 3.6 gives a tighter upper bound for the learning rate than [37]. We require $\eta^{(t)} \le \frac{2}{C K \alpha_j^{(t)}}$ for all $j \in [K]$, compared to $min_{j \in [K]} \frac{1}{C K \alpha_j^{(t)}}$ in [Theorem 5.4, 37]. Our bound is better with multiplicative constant 2. See Line 710-712 for details.
>  - For meta-learning, Thm 3.7 is a novel result of the NBS on the monotonicity of validation loss, which has not been presented before as [37] did not focus on meta-learning. In fact, we generalized [Lemma 1, 43] from a fixed protocol to any protocol $\beta$ with the finite norm. Setting the bargained $\alpha$ to $\beta$ gives a new desirable property of the NBS in mete-learning, which establishes its validity as a meta-learning protocol.
>  - Compared to the setup in [37], we also provided a careful characterization of the feasible set $A$ and extended the discussion from a single disagreement point $D=0$ to general $D$ in Appendix A.4 (Line 611-643). This may provide directions for future gradient aggregation work (Line 350-353).
> We updated our manuscript with the aforementioned comparison.
>
> **Q2.2: Experimental results matching previous works.**
>
> Our experimental results on real data majorly agree with the previous results. We've carefully characterized the pros and cons of previous methods and the effect of bargaining on different previous methods, along with their comparisons in Sections 2 and 4.2. We've updated our manuscript to include this point accordingly.

---

> ### Comment · Reviewer_RvEe · 2024-08-07
>
> Thanks for the response and clarifications. I would recommend that the new contributions be better highlighted to make them clear, and that the other changes be included as well.

---

> > ### Author Response · Authors · 2024-08-08
> >
> > Of course! We will highlight the theoretical contributions and incorporate other changes as suggested. Thank you for helping us to improve the paper!

---

### Official Review · Reviewer_hQxX · 2024-07-12

**Soundness:** 3
**Presentation:** 3
**Contribution:** 3
**Rating:** 7
**Confidence:** 3

**Summary:**

This paper studies fair prediction tasks where fairness is defined on some partition of the data points into groups (by gender or race, for examples). The paper studies a meta-learning framework that only needs access to group labels for the validation set rather than the entire training dataset. An outer hypergradient optimization of minibatch-level example weights is used to optimize for fairness (on the validation loss), interwoven with standard gradient optimization of the model parameters for the typical minibatch loss (given the weights).

Previous work has deployed this approach with various predetermined weighting schemes for the outer optimization. The current work identifies a challenge with the stability of such approaches — in particular, often during an outer optimization there may exist a group whose validation loss hypergradient is not aligned with the overall hypergradient, leading to potentially unstable learning and failure to converge to the Pareto front (that is, the undominated groupies loss frontier). To address this challenge, the current work proposes a first stage (conducted for several epochs) wherein the outer optimization is conducted by Nash bargaining between the groups: Roughly speaking, a hypergradient is selected which maximizes the product of alignment between the groups, relative to a default alternative of no change to the re-weighing hyperparameters. This is done for some time, ideally until the agents are able to converge to the Pareto front, at which time a secondary optimization for a particular fairness objective similar to prior work is pursued as a fine-tuning.

The method is validated in theory by demonstrating Pareto improvement under smoothness and boundedness conditions. Also, experiments are conducted on synthetic and real-world data. The benefit of the synthetic experiments is that the Pareto front can be explicitly computed, and that the method succeeds in converging to the front. The real world experiments show some improvements, though not universal or always dramatic, in fairness for predictive tasks including with unbalanced data, compared to previous meta-learning approaches that do not utilize Nash Bargaining.

**Strengths:**

The paper does a good job of identifying a serious challenge to the learning stability of prior meta learning techniques for fairness. The challenges of hypergradient conflict is intuitive, coherent, and well described. The methods proposed, inspired by Nash Bargaining, are reasonable and original for resolving these conflicts.

The theoretical work is a nice contribution, providing both closed form solutions for the bargaining step and a well-grounded argument for Pareto improvement throughout the process.

The paper does a good job of presenting a lot of results in a clear way with attention placed on visualizations and comprehensive tables for the benefit of the reader.

Several different datasets are used and presented to do a good job of not overstating performance based on a single dataset.

In terms of impact and significance, group fairness-aware predictive models are of clear importance to the community, and the paper addresses one promising technique for dealing with the difficult case where group labels are not generally available for the entire training dataset. The technique proposed seems promising for continued development in contexts of fairness as well as simple problems of class imbalance.

**Weaknesses:**

Lots of empirical results and observations on synthetic data and for a particular model architecture, but any specifics about the synthetic data and model architecture are hidden in the appendix. Of particular concern is the lack of discussion around limitations of the scale of the experiments, given that the proposed method introduces additional complexity into the training process designed to address instability of existing meta learning approaches. The real-world dataset examples use very small MLPs with a single 128 neuron hidden layer, and the image examples from CIFAR only use 5,000 or 500 training examples. Similarly, the real-world experiments use (I believe) at most 5 groups.

Discussion around line 275 “Our experiments show…does not deviate the model from the Pareto front…” I think it should be noted explicitly and emphasized that these experiments are on synthetic examples constructed for purpose of simplicity of analysis, so this trend should not be taken as a given in real-world data where one cannot necessarily even calculate the Pareto front to be able to verify the property.

It would be nice to see some evaluation of the frequency with which bargaining fails in the real-world data, as well as the relative importance for resolving hypergradient conflicts between (1) constraining optimization to the agreement set A, and (2) optimizing within A particularly along the Nash objective.

Typos/grammar:
1. “hypergradeint” instead of “hypergradient” on line 91.
2. Line 144-145, found the phrasing a little confusing
3. Line 271 “if encountered one”

**Questions:**

Q1. Referring to the discussion around lines 151-160. What is the relative significance for the empirical optimization results for simply constraining the hypergradient optimization to the alternative set A [requiring improvement for all groups] versus the selection of the Nash objective in particular to optimize within A?

Q2. Also, how common is it for A to be empty, and would you expect this to become increasingly the case when looking at larger numbers of groups? It seems that in the experiments (referring to the Appendix A.6 details) there are only ever <= 5 groups, is it possible that this approach could stall or encounter more challenges for larger numbers of groups?

**Limitations:**

No concern

---

> ### Author Rebuttal · Authors · 2024-08-04
>
> We deeply appreciate the reviewer’s thoughtful feedback and recognition of our contributions. Thank you for giving us the opportunity to clarify our approach based on your insightful comments.
>
> ---
> **W1: Discussion on the scale of experiments.**
>
> Thank you for your comment! We've updated Section 4.2 to include the potential limitations of the scale of experiments (in particular, on the effect of model size, training size, and number of groups, as suggested). Though we do not extensively explore very large-scale problems due to the scope of this paper, we encourage future studies in this direction.
>
>
> ---
> **W2: Emphasis on synthetic experiments.**
>
> Thank you for pointing it out! We agree and have changed the phrasing to ``experiments on synthetic data" in Line 275 and all other places (Line 108, 574, etc) to avoid overstating the observed properties from simulations. We've also added a clarification on real-world data as suggested.
>
>
> ---
> **W3 / Q1: Constraining optimization to set $A$.**
>
> Thank you for your question! Appendix A.2 gives the comparison with conflict resolution methods, which attempt to give more aligned updates but do not gaurentee to be within $A$. To strictly constrain a proposed update to a non-empty $A$, there are two potential ways as follows:
>
>    1. Choose one update from $A$. This is essentially a bargaining process (where players choose feasible points from $A$). First, a random choice from $A$ would lose the axiomatic property of Pareto Optimality and may result in performance degradation. Second, for axiomatic choices (i.e. solutions characterized by their axioms), we prefer the Nash solution among other prominent ones (including the Kalai-Smorodinsky solution and the Egalitarian solution). Specifically, the Nash objective aligns with our objective to maximize the joint in-effect update. It stands out for its general applicability, robustness [34], and as the unique solution satisfying desirable axioms.
>
>    2. Transform the update from previous weighting protocols (LtR, FORML, Meta-gDRO) to $A$. As far as we understand, there is no simple way to project or rescale the proposed update into $A$.
>
> We've updated our manuscript to include this discussion accordingly.
>
> ___
> **Q2.1: Feasibility of $A$.**
>
> Thank you for your question! The following table indicates the number of non-empty $A$ in the first 15 epochs on the US Crime dataset as a showcase.
>
> |                        | LtR          | FORML       | Meta-gDRO    |
> |------------------------|--------------|-------------|--------------|
> | Hypergradients Aligned | 759 (59.5%)  | 0 (0%)      | 365 (28.6%)  |
> | $A$ Non-empty          | 1046 (82.0%) | 974 (76.4%) | 1072 (84.1%) |
>
> Here, "hypergradients aligned" means the proposed update of the mini-batch lies in $A$ ($\nabla L_{\beta}^\top g_i > 0$ for all $i$). The percentage in parenthesis is the portion out of the total number of 1275 mini-batches. We observed that approximately 80% times the $A$ is nonempty, which gives room for improvement with bargaining. We've updated our manuscript accordingly.
>
>
> **Q2.2: Scalability with larger number of groups.**
>
> Although the likelihood of getting unresolvable conflict may get higher due to the fact that more players are getting involved, we think that the feasibility of $A$ depends more on the group structure rather than the number of groups. For example, the interdependencies and shared factors between groups may cause dependency in hypergradients to enable the feasibility of $A$. When goals of groups rely on common resources or have shared objectives at a higher level, it is still likely to have nonempty $A$.
>
> Theoretically, all presented results should still hold as the number of group increases, thanks to the novel techniques employed in Theorem 3.2 for making this possible. For exmaple, if the number of subgroups exceeds the dimension of the hypergradient, it’s impossible to have linear independence and the previous result [Theorem 5.4, 37] cannot apply.
>
> We've updated our manuscript to include this discussion.
>
> ___
> **W4: Typos.**
>
> Thanks for pointing them out! We've fixed the typos and improved the phrasing on Line 144-145 as suggested.

---

> > ### Comment · Reviewer_hQxX · 2024-08-10
> >
> > I acknowledge that I have read the author rebuttal. Thank you very much for your detailed responses. I hope that the discussion has improved the paper and appreciate the contributions of the current work even if there are interesting questions of scale for the future. A nice paper, in my opinion!

---

> > > ### Author Response · Authors · 2024-08-12
> > >
> > > Thank you so much for your kind words! We agree and appreciate your feedback. We’ve been making editions according to the comments along the way and will finalize in our camera-ready version.

---

### Official Review · Reviewer_gYXi · 2024-07-13

**Soundness:** 4
**Presentation:** 3
**Contribution:** 3
**Rating:** 6
**Confidence:** 2

**Summary:**

This paper proposes a novel method to solve the unfairness issue in machine learning. In particular, the authors observed that, existing methods may cause "hypergradient conflict" during the optimization process. To resolve the hypergradient conflict, this paper applies the Nash bargaining solution (NBS), and operates in two stages: The first stage resolves the hypergradient conflict and obtains a solution near the Pareto front, and the next stage applies the fairness constraints.

**Strengths:**

1. This paper presents a subtle yet important observation on hypergradient conflict, that can cause inefficiency while learning fair representations.

2. The application of the NBS is novel and seems appropriate for this scenario.

3. The algorithm is provided with extensive theoretical justifications (section 3.5).

**Weaknesses:**

Although the approach is novel, more technical explanations on the game theoretical model may be needed. In particular, as far as I understand the work, incentives play a critical role and they provide justifications of many important processes, e.g. the bargaining. However, the origins of payoffs, the set of feasible utility payoffs, and disagreement point (on Page 4) were not emphasized, so I am interested in how the payoffs are determined? Also, were they pre-determined and fixed, or do they change during the algorithm?

**Questions:**

In equation (4), the utility is defined as $u_i (\nabla L_{\alpha}) = g_i^{\top} \nabla L_{\alpha}$. It is easy to see that if the value is non-positive, then there is a misalignment. However, if it is positive, does the magnitude tell us some information, e.g. if the value is large?

**Limitations:**

Yes, future directions are discussed at the last section.

---

> ### Author Rebuttal · Authors · 2024-08-04
>
> Thank you for your insightful feedback and constructive comments. We are grateful for the chance to discuss these questions.
>
> **W1: Technical explanations on the game theoretical model.**
>
> Thank you for your comment! The concepts you mentioned lacking emphasis on Page 4 have their detailed definition subsequently on Page 5 (Line 151-160). Additional discussion on how incentives play roles is in Section 3.3. We also supplemented further explanations on the problem setup in Appendix A.4, the technical preliminaries in Appendix A.3, and the game theory context in Appendix A.2. Please let us know if you have further questions.
>
> **W2: How payoffs are determined?**
>
> Thank you for your question! Equation 4 gives the definition of payoffs (i.e. utilities). Given a proposed update, the payoff of one group is determined by the dot product between the proposed update and the hypergradient of this group. Though the way to compute payoff is pre-determined, the values of payoffs change during the algorithm and serve as a criterion to compute the bargained update.
>
> **Q1: Magnitude of payoff.**
>
> Thank you for your question! The value of utility tells us how much of the proposed update is applied in the direction of hypergradient of a certain group. Informally, if it is positive and the magnitude is large, it means the "in-effect" update for such group is large and this group is likely to be improved much by the proposed update (vice versa). We refer to Line 151-160 for detailed explanations.

---

> > ### Comment · Reviewer_gYXi · 2024-08-13
> > **Thanks for the response**
> >
> > I appreciate the authors' response, which addresses my concern. I have decided to raise my rating from 5 to 6.

---

> > > ### Author Response · Authors · 2024-08-13
> > >
> > > Thank you for your time! We’re glad that our responses addressed your concern, and we appreciate your interest and consideration!

---

### Official Review · Reviewer_YN1k · 2024-07-20

**Soundness:** 3
**Presentation:** 3
**Contribution:** 3
**Rating:** 6
**Confidence:** 4

**Summary:**

The paper addresses group-level fairness in ML with two-stage meta learning. The modeler is given access to a sensitive-group-labeled *validation* dataset and must simultaneously design how to (linearly) weight each group loss in validation via bargaining to resolve gradient conflicts, and how to translate those group weights into weights for each unlabeled sample in the training set (minibatch).

The paper is supported by a proof of their Nash bargaining solution that does not rely on linear independence and empirical results for their implementation.

**Strengths:**

The NBS solution proposed here is very well motivated and, to my knowledge, novel in this particular sub-field. The overall goal of the NBS is to find optimal weights \beta for each group loss. The utility per group is computed as the inner product of the gradient wrt the group loss and the grad wrt the weighted group loss function; the disagreement payoff for each player (group) is to simply stop optimization and keep the model as-is. Essentially, group weights are negotiated such that each group's loss gradient has sufficient (positive) alignment, and axiomatically disallows any one group to be unilaterally overruled (the 'do nothing' disagreement outcome has a higher overall payoff than any solution where any player sees their utility alignment become negative).

The resulting algorithm is relatively straightforward, though compared to simpler two-stage approaches, the bargaining stage requires K second order gradients per minibatch, as well as solving for a non-linear, non-negative equation to devise the optimal weights alpha per bargaining round

**Weaknesses:**

The computational costs of the bargaining phase could be considerable (see strengths). To this extent, the authors actually only run the bargaining stage in some predefined set of bargaining rounds Tbar and then continue with the weights fixed.

I did not see anything to suggest these weights would remain constant once a bargaining solution is achieved, or any particular guidance or intuition on how this Tbar parameter is selected.

**Questions:**

The experimental section enhances several approaches (LtR, FORML, Meta-gDRO) with the proposed bargaining stage. To my understanding, these all differ only in their original choice of per-group weights \beta and do not incorporate any additional steps in the bargaining phase to ensure the original objective is being preserved.

The results of the enhanced algorithm for all of these experiments are quite different so this would suggest quite a strong sensibility of the overall bargaining procedure to the initial weight \beta^0. I would like the authors to elaborate on this further and provide an intuitive or formal explanation on why the bargaining solutions exhibit such strong sensitivity to this initial parameter.

---

> ### Author Rebuttal · Authors · 2024-08-04
>
> We sincerely thank the reviewer for their thoughtful comments and valuable feedback. We appreciate the opportunity to address these points and clarify our approach.
>
>
> **W1: Computational cost.**
>
> We appreciate this insightful question. We agree the bargaining phase introduces additional complexity to the training process, primarily due to solving the optimization problem in each step. Our analysis shows that the entire training process takes approximately 1.2-1.4 times as long as training with bargaining does, with the bargaining steps (first 15 epochs) themselves taking 2-10 times as much regular training takes per epoch. We view this as a worthwhile trade-off given the enhanced performance and fairness achieved. We've updated our manuscript accordingly.
>
>
>
> **W2: Switching phases and selection of $T_{bar}$.**
>
> Thank you for your question! We refer to our observations from synthetic experiments (Figure 3) to justify the reason to remain a fixed weighting protocol after bargaining: After using bargaining to reach the Pareto Front, switching objective does not diverge the model from the Pareto Front in simulations. This implies that achieving downstream fairness goals may not be at the cost of compromising overall model performance if the model has been converged to the Pareto Front.
>
> In practice, we determine $T_{bar}$ by monitoring the bargaining success rate. We observed from real-data experiments that model's performance would stabilize when this rate stabilizes. This may serve as a sign of swtiching to Stage 2 because no significant improvements could be brought by bargaining. In our real-data experiments, we set $T_{bar}$ to 15 epochs, as this allowed all evaluated settings to reach a stable bargaining success rate. We've added clarifications in our manuscript.
>
> **Q1: Sensitivity to the initial weighting protocol  $\beta_0$.**
>
> We agree and appreciate your acute observation! First, we want to clarify that $\beta_0$ is also used in Stage 1 when bargaining fails, which would preserve the fairness objectives meanwhile providing a fresh start for subsequent bargaining. Second, for the exhibited sensitivity, our analysis to Figure 4 demonstrates that the improvement from bargaining correlates with the initial hypergradient alignment rate (the portion of aligned batches). When this initial rate is low, the bargaining process yields significant improvements (for example, FORML). Conversely, when the initial alignment rate is high, the gains from bargaining are more modest. This relationship may provide insight into the varying effectiveness of our approach across different scenarios.

---

> > ### Comment · Reviewer_YN1k · 2024-08-12
> >
> > Thanks for the clarifications. I read the other reviewers' comments and still like the paper, so I'll continue to advocate for its acceptance (with my current score).

---

> > > ### Author Response · Authors · 2024-08-12
> > >
> > > Thank you so much for your advocacy! We’re glad that you like our paper. We’ve been working on improving the paper for the camera-ready version as suggested. Your time and review is greatly appreciated.

---

### Decision · Program_Chairs · 2024-09-25

**Decision:**

Accept (poster)

**Comment:**

The paper studies imposing group fairness via two-stage meta-learning. The authors observe hypergradient conflicts (i.e., a group's hypergradient is in conflict with the overall hypergradient) as a problem during the optimization for group fairnes via meta-learning. To resolve this problem, the authors propose a two stage process: In the first stage, a hypergradient is selected using ideas from Nash Bargarining Solution with the goal of converging to a Pareto frontier. In the second stage, a particular fairness objective is persued (which is similar to prior work).

The reviewer liked this work and in particular, both the observation about hypergradient conflict and using Nash Bargaining Solution were well received. The reviewers, however, had some concerns. The first one is the scalability of the bargaining step to larger number of groups, bigger datasets, and bigger models. The second concern is the selection of $T_{\text{bar}}$ and switching to the second stage. The reviewers found the authors' response to both these questions satisfying. So I encourage the authors to incorporate the responses into the final version even if the response indicates limitations of the work (e.g., scaling).